# Mapping the drivers of formaldehyde (HCHO) variability from 2015 to 2019 over eastern China: insights from FTIR observation and GEOS-Chem model simulation

Youwen Sun [1], Hao Yin [1, 3], Cheng Liu [2, 3, 8, 9, 1]*, Lin Zhang [4]*, Yuan Cheng [5]*, Mathias Palm [6], Justus Notholt [6], Xiao Lu [7], Corinne Vigouroux [10], Bo Zheng [11], Wei Wang [1], Nicholas Jones [12], Changong Shan [1], Min Qin[1], Yuan, Tian [13], Qihou Hu [1], and Jianguo Liu [1]

(1 *Key Laboratory of Environmental Optics and Technology, Anhui Institute of Optics and Fine Mechanics, HFIPS, Chinese Academy of Sciences, Hefei 230031, China*)

(2 *Center for Excellence in Regional Atmospheric Environment, Institute of Urban Environment, Chinese Academy of Sciences, Xiamen, 361021, China*)

(3 *Department of Precision Machinery and Precision Instrumentation, University of Science and Technology of China, Hefei, 230026, China*)

(4 *Laboratory for Climate and Ocean–Atmosphere Studies, Department of Atmospheric and Oceanic Sciences, School of Physics, Peking University, Beijing 100871, China*)

(5 *State Key Laboratory of Urban Water Resource and Environment, School of Environment, Harbin Institute of Technology, Harbin 150090, China*)

(6 *Institute of Environmental Physics, University of Bremen, P. O. Box 330440, 28334 Bremen, Germany*)

(7 *School of Engineering and Applied Sciences, Harvard University, Cambridge, MA 02138, USA*)

(8 *Key Laboratory of Precision Scientific Instrumentation of Anhui Higher Education Institutes, University of Science and Technology of China, Hefei, 230026, China*)

(9 *Anhui Province Key Laboratory of Polar Environment and Global Change, University of Science and Technology of China, Hefei, 230026, China*)

(10 *Royal Belgian Institute for Space Aeronomy (BIRA-IASB), Brussels, Belgium*)

(11 *Laboratoire des Sciences du Climat et de l'Environnement, CEA-CNRS-UVSQ, UMR8212, Gif-sur-Yvette, France*)

(12 *School of Chemistry, University of Wollongong, Northfields Ave, Wollongong, NSW, 2522, Australia*)

(13 *Anhui University Institutes of Physical Science and Information Technology, Hefei 230601, China*)

Correspondence: Cheng Liu (chliu81@ustc.edu.cn), Lin Zhang (zhanglg@pku.edu.cn) and Yuan Cheng (ycheng@hit.edu.cn)

**Abstract:**

The major air pollutant emissions have decreased and the overall air quality has substantially improved across China in recent years as a consequence of active clean air policies for mitigating severe air pollution problems. As key precursors of formaldehyde (HCHO) and ozone ($O_3$), the volatile organic compounds (VOCs) in China are still increasing due to the lack of mitigation measures for VOCs. In this study, we investigated the drivers of HCHO variability from 2015 to 2019 over Hefei, eastern China by using ground-based high-resolution Fourier transform infrared (FTIR) spectroscopy and GEOS-Chem model simulations. Seasonal and interannual variabilities of HCHO over Hefei were analysed and hydroxyl (OH) radical production rates from HCHO photolysis were evaluated. The relative contributions of emitted and photochemical sources to the

observed HCHO were analysed by using ground level carbon monoxide (CO) and $O_x$ ($O_3$ + nitrogen oxide ($NO_2$)) as tracers for emitted and photochemical HCHO, respectively. Contributions of emission sources from various categories and geographical regions to the observed HCHO summertime enhancements were determined by using a series of GEOS-Chem sensitivity simulations. The column-averaged dry air mole fractions of HCHO ($X_{HCHO}$) reached a maximum monthly mean value of $1.1 \pm 0.27$ ppbv in July and a minimum monthly mean value of $0.4 \pm 0.11$ ppbv in January. The $X_{HCHO}$ time series from 2015 to 2019 over Hefei showed a positive change rate of $2.38 \pm 0.71$ % per year. The photochemical HCHO is the dominant source of atmospheric HCHO over Hefei for most of the year (68.1%). In the studied years, the HCHO photolysis was an important source of OH radical over Hefei during all sunlight hours of both summer and winter days. The oxidations of both methane ($CH_4$) and nonmethane VOCs (NMVOCs) dominate the HCHO production over Hefei and constitute the main driver of its summertime enhancements. The NMVOCs related HCHO summertime enhancements were dominated by the emissions within eastern China. The observed increasing change rate of HCHO from 2015 to 2019 over Hefei was attributed to the increase in photochemical HCHO resulting from increasing change rates of both $CH_4$ and NMVOCs oxidations, which overwhelmed the decrease in emitted HCHO. This study provides a valuable evaluation of recent VOCs emissions and regional photochemical capacity in China. In addition, understanding the sources of HCHO is a necessary step for tackling air pollution in eastern China and mitigating the emissions of pollutants.

**1 Introduction**

Formaldehyde (HCHO) is one of the most critical tropospheric pollutants, which not only directly threatens human health but also plays a significant role in atmospheric photochemical reactions (Franco et al., 2015; Jones et al., 2009; Notholt et al., 1992; Notholt et al., 2000; Vigouroux et al., 2009). Indeed, the chemistry of HCHO is common to virtually all mechanisms of tropospheric chemistry (Chapter 6, Seinfeld and Pandis, 2016). Furthermore, the observation of HCHO variability allows us to constrain volatile organic compounds (VOCs) emissions and improve current understanding of the complex degradation mechanisms of VOCs (e.g. Palmer et al., 2003; Millet et al., 2008; Boersma et al., 2009; Stavrakou et al., 2009; Fortems-Cheiney et al., 2012; Barkley et al., 2013; Marais et al., 2014; Streets et al., 2013; Gao et al., 2018).

Natural source such as biomass burning emission and anthropogenic sources such as vehicle exhausts, industrial emissions, and coal combustions can emit HCHO directly into the atmosphere (Albrecht et al., 2002; Holzinger et al., 1999). The emitted HCHO is mainly attributed to incomplete combustion and is closely associated with the emissions of benzene ($C_9H_{12}O$), toluene ($C_7H_8$), or carbon monoxide (CO) (Friedfeld et al., 2002; Garcia et al. 2006; Ma et al., 2016). In addition, photochemical formation of HCHO has been identified from the atmospheric oxidation of methane ($CH_4$) and numerous nonmethane VOCs (NMVOCs), which are closely associated with the formation of ozone ($O_3$) or $O_x$ ($O_3$ + nitrogen dioxide ($NO_2$)) or glyoxal (CHOCHO) (Chapter 6, Seinfeld and Pandis, 2016; Friedfeld et al., 2002; Garcia et al. 2006; Zhang et al., 2020). As a result, the relative contribution of emitted and photochemical sources to atmospheric HCHO can be estimated via a linear multiple regression analysis method that aims at reproducing the time series of observed HCHO from a linear combination of the time series of CO (or $C_9H_{12}O$ or $C_7H_8$ ) and $O_3$ (or $O_x$ or CHOCHO) as tracers for emitted and photochemical HCHO, respectively (Friedfeld et al., 2002; Garcia et al., 2006; Hong et al., 2018; Li et al., 2010; Lui et al., 2017; Ma et al., 2016; Su

et al., 2019). The separation between emitted and photochemical sources of HCHO is important for improving the air quality in megacities (Garcia et al., 2006).

The relative contribution of emitted and photochemical sources to atmospheric HCHO has been analysed by using the CO–O$_3$ (Friedfeld et al., 2002; Li et al., 2010; Lui et al., 2017; Su et al., 2019), CO–O$_x$ (Hong et al., 2018), CO–CHOCHO (Garcia et al., 2006) and CO/C$_9$H$_{12}$O/C$_7$H$_8$–O$_3$ (Ma et al., 2016) tracer pairs in various polluted environments. In these studies, HCHO column measurements were sometimes used as representative of near-surface conditions because the HCHO has a vertical distribution that is heavily weighted toward the lower troposphere over polluted areas. Improved knowledge of the contributions of different emission categories and geographical regions to HCHO enhancements is significant for improving the understanding of the HCHO production regime, and further for regulatory and control purposes (Molina and Molina, 2002; Surl et al., 2018). However, previous studies have often concentrated on the separation between emitted and photochemical sources of HCHO, while contributions of different emission categories and geographical regions were rarely mentioned or only analysed qualitatively by using the back-trajectories analysis technique (Friedfeld et al., 2002; Franco et al., 2016; Garcia et al., 2006; Hong et al., 2018; Li et al., 2010; Lui et al., 2017; Ma et al., 2016; Su et al., 2019). In this study, the drivers of HCHO variability over Hefei, eastern China were mapped using ground-based high-resolution Fourier transform infrared (FTIR) spectroscopy and GEOS-Chem model simulation. Seasonal and interannual variabilities of HCHO were investigated and hydroxyl (OH) radical production rates from HCHO were evaluated. In addition to separation between emitted and photochemical sources, contributions of different emission categories and geographical regions to the observed HCHO summertime enhancement were also investigated.

China has implemented a series of active clean air policies in recent years to mitigate severe air pollution problems. Therefore, the emissions of major air pollutants have decreased, and the overall air quality has substantially improved (Sun et al., 2019; Zhang et al., 2019; Zheng et al., 2018). However, current clean air policies lack mitigation measures for VOCs, which are key precursors of HCHO and O$_3$ (Lu et al., 2018; Zheng et al., 2018). The recent increasing trend in O$_3$ in China was largely attributed to the increase in VOCs in recent years (Lu et al., 2019). Multi-year time series of ground-based FTIR measurements of HCHO in this study provide an evaluation of recent regional VOCs emissions over eastern China. The degradation of HCHO provides a large source of OH radicals, which play a significant role in atmospheric photochemical reactions (Chapter 6, Seinfeld and Pandis, 2016). The OH radical production rates from HCHO photolysis estimated in this study provide an evaluation of regional photochemical capacity related to the degradation of HCHO over eastern China. In addition, understanding the sources of HCHO is a necessary step for tackling the problems of poor air quality in eastern China and mitigating the emissions of pollutants.

The next section describes the methodology which includes site description and instrumentation, the ground-based FTIR HCHO dataset, the 3$^{rd}$ regression model used to determine seasonal and interannual variabilities of HCHO, the linear regression model used for source separation, and the GEOS-Chem sensitivity simulations used for source attribution. Section 3 reports the results for comparison with ground level in situ measurements, HCHO variability on different time scales, source separation, and OH radical production rates from HCHO photolysis. Section 4 reports the results for source attribution using GEOS-Chem sensitivity simulations. Conclusions are presented in section 5.

## 2 Methodology

### 2.1 Site description and instrumentation

As shown in Fig.1, the FTIR observation site (117°10′E, 31°54′N, 30 m a.s.l. (above sea level)) is located on an island in the western suburbs of the megacity Hefei (the capital of Anhui Province) in eastern China (Tian et al., 2017). The Anhui Institute of Optics and Fine Mechanics, Chinese Academy of Sciences (AIOFM-CAS) directly operates this site on campus, adjacent to the Shu Shan Lake that covers an area of 207.5 km$^2$. This area prevails southeast winds in summer and northwest winds in winter. The regional landscape is mostly flat with a few hills. The downtown Hefei is located to the southeast of this site and is densely populated with seven million people. The site is surrounded by wetlands or cultivated lands in other directions. Local anthropogenic emissions mainly come from the city and natural emissions are originated from cultivated lands or wetlands.

The FTIR observatory consists of a high resolution FTIR spectrometer (IFS125HR, Bruker GmbH, Germany) and a solar tracker (Tracker-A Solar 547, Bruker GmbH, Germany). The near infrared (NIR) and middle infrared (MIR) solar spectra are alternately recorded in routine observations (Wang et al., 2017). The MIR spectra are recorded with a spectral resolution of 0.005cm$^{-1}$ which follows the requirements of Network for Detection of Atmospheric Composition Change (NDACC, http://www.ndacc.org/, last accessed on 3 June 2019). In this study, the instrument is equipped with a KBr beam splitter, an InSb detector, and a filter centered at 2800 cm$^{-1}$ for HCHO measurements. The entrance field stop size ranging from 0.80 to 1.5 mm was employed to maximise the signal to noise ratio (SNR) consistent with the maximum frequency possible for the selected wavenumber range. The number of HCHO measurements on each measurement day varied from 1 to 17 with an average of 6. In total, there were 523 days of qualified measurements between 2015 and 2019.

Ground level hourly mean concentrations of CO, $O_3$, and $NO_2$ from 2015 to 2019 were provided by the China National Environmental Monitoring Center (CNEMC) network operated by the Chinese Ministry of Ecology and Environment (http://www.cnemc.cn/en/, last access: 22 March 2020). The CNEMC network has monitored six ground level air pollutants (including CO, $O_3$, $NO_2$, $SO_2$, $PM_{10}$, and $PM_{2.5}$) in nationwide sites in mainland China since 2013, and by 2019, it was extended to more than 2000 sites in 454 cities. The datasets have been used in many studies to evaluate local air quality over China (Gao et al., 2018; Hong et al., 2018; Hu et al., 2017; Li et al., 2018; Lu et al., 2018; Shen et al., 2019; Su et al., 2019). The measurements taken at the nearest CNEMC site were used in this study, which is approximately 700 m away from the FTIR site (Fig. 1(b)). The $O_3$ and $NO_2$ measurements are based on a differential optical absorption spectroscopy (DOAS) analyser and the CO measurements are based on a gas correlation filter infrared analyser. All analysers are regularly calibrated by the CNEMC staffs to ensure the measurement errors for all gases are within 2% (http://www.cnemc.cn/en/, last access: 22 March 2020).

Ground level 10-minute mean concentrations of HCHO from 2017 to 2019 were provided by a long-path DOAS instrument (LP-DOAS) located over approximately 900 m away from the FTIR site (Fig. 1(b)). The LP-DOAS instrument consists of a 150 W xenon short-arc lamp as the light source, a telescope with diameter of 220 mm and focal length of 650 mm acting as the transmitting and receiving component, a retro-reflector and a grating spectrograph. The telescope and the retro-reflector were placed about 30 m above ground at two buildings which are separated by a distance of 350 m, resulting in a total measurement optical path of 700 m. Light from xenon lamp is directed to the telescope and transmitted towards to the retro-reflector. Reflected light from the retro-reflector

is received by the same telescope and redirected to the spectrograph for spectral analysis. A fan is
installed in the emitter/receiver unit to avoid the influence of $O_3$ generated by the xenon lamp. A
similar experimental setup with LEDs as the light source have been demonstrated by Chan et al.
(2012) and Zheng et al. (2018). The measurement error for HCHO by the LP-DOAS is estimated to
be 3% (Chan et al., 2012; Ling et al., 2014).
Ground level minutely mean concentrations of $H_2O$ from 2015 to 2019 were available from a
cavity ring-down spectroscopy (CRDS) analyser (G2401m, Picarro, Inc., USA) which is located
side by side with the FTIR spectrometer (Fig. 1(b)). The CRDS analyzer samples ambient air on the
building roof near the solar tracker dome and outputs $H_2O$ concentration with a measurement error
of 1%.
**2.2 Ground-based FTIR HCHO dataset**
**2.2.1 Retrieval strategy**
The SFIT4 version 0.9.4.4 algorithm was used in the HCHO retrieval, and the retrieval settings
were prescribed from a harmonisation project described in Vigouroux et al. (2018). We refer to
Pougatchev et al. (1995) for more details on the retrieval principles. Total columns and volume
mixing ratio (VMR) vertical profiles of HCHO are obtained from its pressure and temperature
dependent absorption lines. Four micro-windows (MWs) were used: two strong lines within 2778–
2782 cm$^{-1}$ and two weak lines within 2763 – 2766 cm$^{-1}$ (Vigouroux et al., 2018). The profiles of
$CH_4$ and $O_3$ and total columns of HDO and $N_2O$ were simultaneously retrieved in addition to the
HCHO profile for minimising the cross interference.
All spectroscopic absorption parameters were based on the atm16 line list from the compilation
of Geoffrey Toon (Vigouroux et al., 2018). In this atm16 line list, the HCHO and $N_2O$ lines
correspond to the HITRAN 2012 database (Rothman et al., 2013). This HITRAN 2012 database
includes the latest improved HCHO parameters (broadening coefficients, Jacquemart et al., 2010),
which complement the release in HITRAN 2008 (Rothman et al., 2009) of new HCHO line
intensities from the same group (Perrin et al., 2009). The spectroscopic parameters for the lines of
$H_2O$ and its isotopologues in atm16 are from Toth 2003 (http://mark4sun.jpl.nasa.gov/data/spec/
H2O/RAToth_H2O.tar; last access: 5 September 2019); some lines from $O_3$ and $CH_4$ in the vicinity
of HCHO have been empirically adjusted or replaced with older HITRAN versions in atm16 when
obvious problems were found in the HITRAN 2012 database (Vigouroux et al., 2018).
The *a priori* profiles of temperature, pressure, and $H_2O$ were interpolated from the National
Centers for Environmental Protection (NCEP) 6-hourly reanalysis data and the *a priori* profiles of
other gases were from the averages of the Whole-Atmosphere Community Climate Model version
6 (WACCM) simulations from 1980 to 2020. The diagonal elements of the *a priori* profile
covariance matrix $\mathbf{S}_a$ and the measurement noise covariance matrix $\mathbf{S}_\varepsilon$ were set to standard deviation
(SD) of the WACCM simulations and the inverse square of the signal-to-noise ratio (SNR) of each
spectrum, respectively. The non-diagonal elements of both $\mathbf{S}_a$ and $\mathbf{S}_\varepsilon$ were set to zero. We regularly
used a low-pressure HBr cell to diagnose the instrument line shape (ILS) of the high resolution FTIR
spectrometer at Hefei and included the measured ILS in the retrieval (Hase et al., 2012; Sun et al.,
40 2018).

**2.2.2 Averaging kernels and error budget**
The vertical information contained in the FTIR retrievals can be characterized by the averaging

kernel matrix **A** (Rodgers, 2000). The rows of **A** are the so called averaging kernels and they represent the sensitivity of the retrieved profile to the real profile. Their full width at half maximum (FWHM) is a measure of the vertical resolution of the retrieval at a given altitude. The area of averaging kernels represents sensitivity of the retrievals to the measurement. This sensitivity at a specific altitude is calculated as the sum of the elements of the corresponding averaging kernels (Vigouroux et al., 2018). It indicates, at each altitude, the fraction of the retrieval at each altitude that comes from the measurement rather than from the *a priori* information (Rodgers, 2000). A value close to zero at a certain altitude indicates that the retrieved profile at that altitude is nearly independent of measurement and is therefore approaching the *a priori* profile. The trace of the averaging kernel matrix **A** is the so called degrees of freedom for signal (DOFS) and it quantifies the number of independent information in the retrieved vertical profile.

Fig. 2 shows the averaging kernels, cumulative sum of DOFS, and VMR profile of randomly selected HCHO retrieval at Hefei. The ground-based FTIR measurements of HCHO at Hefei have a sensitivity larger than 0.5 from the ground to about 15 km altitude, indicating that the retrievals are mainly sensitive to the troposphere. This also means that the retrieved profile information above 15 km comes for less than 50% from the measurement, or in other words, that the *a priori* information influences the retrieval by more than 50%. The typical DOFS over the total atmosphere obtained at Hefei for HCHO is $1.2 \pm 0.2$ ($1\sigma$), meaning that we cannot provide more than one piece of information on the vertical profile. This is the reason that only total columns of HCHO or column-averaged dry air mole fractions of HCHO ($X_{HCHO}$) are discussed in this paper and not vertical profiles. As expected by the low DOFS, the shape of the retrieved profile which is heavily weighted toward the lower troposphere is very similar to the shape of the *a priori* profile (not shown here).

We calculated the error budget following the formalism of Rodgers (2000), and separated all error items into systematic or random errors depending on whether they are constant over consecutive measurements, or vary randomly. Table 1 summarizes the random, systematic, and the combined error budget for total column of HCHO demonstrated in Fig. 2. The input covariance matrix of temperature has been estimated using the differences between an ensemble of NCEP and Sonde temperature profiles near Hefei, leading to 2 to 5 K in the troposphere and 3 to 7 K in the stratosphere. For each interfering species, the associated covariance matrix were obtained with the WACCM v6 climatology. The input covariance matrix of measurement error are based on the inverse square of the SNR of each spectrum (see section 2.2.1). The FTIR spectrometer at Hefei is assumed to be not far from the ideal condition, and the input uncertainties for background curvature, optical path difference, field of view, interferogram phase, and ILS are estimated to be 1%. For the HCHO spectroscopic parameters, the line list in atm16 follows HITRAN 2012 (Rothman et al., 2013), which used the work of Jacquemart et al. (2010), and we use 10% for line intensity, pressure-, and temperature-broadening coefficients. For each individual retrieval parameter and the smoothing error, the input covariance matrix are prescribed from the optimal estimation retrieval output.

We see from Table 1 that the major random errors for HCHO retrieval at Hefei are measurement noise (1.59%), smoothing error (0.83%), and temperature uncertainty (0.61%), and the major systematic errors are line intensity uncertainty (9.04%) and line pressure broadening uncertainty (6.60%). Total random and systematic errors are estimated to be 1.71% and 11.24%, respectively. Total retrieval error calculated as square root sum of the squares of total random and systematic errors is estimated to be 12.29%.

**2.3 Regression model for seasonal and interannual variabilities**

The seasonality and interannual variability of HCHO from 2015 to 2019 were evaluated using a bootstrap resampling method following that of Gardiner et al. (2008). Gardiner et al. (2008)'s method has been used by many studies to estimate the variability of atmospheric compounds on different time scales (including Gardiner et al., 2008; Jones et al., 2009; Sun et al., 2018; Sun et al., 2020; Tian et al., 2017; Viatte et al., 2014; Vigouroux et al., 2015; Vigouroux et al., 2018; Zeng et al., 2012; Franco et al., 2016). The following nonlinear regression model $Y_{hcho}^{mod1}(t)$ was applied to fit the FTIR time series of HCHO $Y_{hcho}^{meas}(t)$:

$$Y_{hcho}^{meas}(t) = Y_{hcho}^{mod1}(t) + \varepsilon(t) \tag{1}$$

$$Y_{hcho}^{mod1}(t) = A_0 + A_1 t + A_2 \cos\left(\frac{2\pi t}{365}\right) + A_3 \sin\left(\frac{2\pi t}{365}\right) + A_4 \cos\left(\frac{4\pi t}{365}\right) + A_5 \sin\left(\frac{4\pi t}{365}\right) \tag{2}$$

$$d\% = \frac{Y_{hcho}^{meas}(t) - Y_{hcho}^{mod1}(t)}{Y_{hcho}^{mod1}(t)} \times 100 \tag{3}$$

where $A_0$ is the intercept, $A_1$ is the annual growth rate, and $A_1/A_0$ is the interannual change rate discussed below. The $A_2 - A_5$ parameters describe the seasonal cycle, $t$ is the measurement time elapsed since 2015, and $\varepsilon(t)$ represents the residuals between the measurements and the fitting model. Generally, the bootstrap resampling model can't capture the diurnal cycle of an atmospheric compound with a large diurnal variability. In order to minimize this influence, we performed the regression fit on daily mean dataset and incorporated the error arising from the autocorrelation in the residual into the uncertainty in the change rate following the procedure of (Santer et al., 2008). Fractional differences of FTIR HCHO time series relative to their seasonal mean values represented by $Y_{hcho}^{mod1}(t)$ were calculated in equation (3) to analyse seasonal enhancements.

**2.4 Regression model for source separation**

The emitted and photochemical HCHO were separated by using ground level CO and $O_x$ ($O_3$ + $NO_2$) as tracers, respectively. The methodology in this study follows that of Friedfeld et al. (2002), which has been used extensively in source separation for atmospheric HCHO (including Friedfeld et al., 2002; Garcia et al., 2006; Hong et al., 2018; Li et al., 1994; Li et al., 2010, Lui et al., 2017; Ma et al., 2016, Su et al., 2019; Wang et al., 2015). Over polluted atmosphere, $O_3$ reacts with nitric oxide (NO) emitted from vehicle exhaust to form $NO_2$. In this case, $O_x$ ($O_3$ + $NO_2$) is in principle a better surrogate of photochemical processes as it also accounts for titrated $O_3$ (Garcia et al., 2006; Li et al., 1994). Therefore, this study uses $O_x$ as a tracer for photochemical HCHO. A linear regression model was used to establish a link among the time series of HCHO, CO, and $O_x$ (Garcia et al., 2006). The observed HCHO $Y_{hcho}^{meas}(t)$ can be reproduced by the following linear regression model $Y_{hcho}^{mod2}(t)$:

$$Y_{hcho}^{meas}(t) = Y_{hcho}^{mod2}(t) + \varepsilon(t) \tag{4}$$

$$Y_{hcho}^{mod2}(t) = \alpha_0 + \alpha_1 \times Y_{co}^{meas}(t) + \alpha_2 \times Y_{ox}^{meas}(t) \tag{5}$$

where $\alpha_0$, $\alpha_1$, and $\alpha_2$ are coefficients obtained from the multiple linear regression fit. $\alpha_0$ which is neither classified as emitted or photochemical contributions represent the regional HCHO condition in the background atmosphere, $\alpha_1$ is the emission ratio of HCHO with respect to CO, and $\alpha_2$ denotes the portion of HCHO from photochemical production. $\varepsilon(t)$ is the fitting residual, which is assumed to be independent with a constant variance and a mean of zero (Garcia et al., 2006). $Y_{co}^{meas}(t)$ and $Y_{ox}^{meas}(t)$ are ground level VMR time series of CO and $O_x$, respectively. The relative contributions of emitted ($R_e$), photochemical ($R_p$), and background ($R_b$) sources to the observed HCHO can be

calculated by (Friedfeld et al., 2002; Garcia et al., 2006; Hong et al., 2018; Li et al. 2010, Lui et al. 2017; Ma et al., 2016; Su et al., 2019; Wang et al., 2015):

$$R_e = \frac{\alpha_1 \times Y_{co}^{meas}(t)}{\alpha_0 + \alpha_1 \times Y_{co}^{meas}(t) + \alpha_2 \times Y_{ox}^{meas}(t)} \tag{6}$$

$$R_p = \frac{\alpha_2 \times Y_{ox}^{meas}(t)}{\alpha_0 + \alpha_1 \times Y_{co}^{meas}(t) + \alpha_2 \times Y_{ox}^{meas}(t)} \tag{7}$$

$$R_b = \frac{\alpha_0}{\alpha_0 + \alpha_1 \times Y_{co}^{meas}(t) + \alpha_2 \times Y_{ox}^{meas}(t)} \tag{8}$$

To compare the results, the regression analysis in Garcia et al. (2006) was run using subsets of data, which comprise a comparable number of data points for each considered time period. By dividing the data into several periods of interest, it is possible to lower the residual and improve the fitting correlation (Garcia et al., 2006). Garcia et al. (2006) also concluded that the fitting results were more robust by using a real background value to initialize the regression analysis. Generally, this initial background level can be approximated by the measurement in the "clean" atmosphere at a rural site or derived from statistics of previous studies in the studied region (Garcia et al., 2006; Hong et al., 2018; Ma et al., 2016; Su et al., 2019; Wang et al., 2015). The findings of Garcia et al. (2006) has been used by many studies in source separation for atmospheric HCHO (including Hong et al., 2018; Li et al., 2010, Lui et al., 2017; Ma et al., 2016, Su et al., 2019; Wang et al., 2015). For multi-year time series of HCHO in this study, we grouped all measurements by month and performed the regression analysis for source separation on a monthly basis. The empirical background level of previous studies in the Yangtze River Delta (YRD) region was used to initialize the regression analysis.

**2.5 GEOS-Chem sensitivity simulations**

**2.5.1 GEOS-Chem model description**

The drivers of the observed HCHO variability were determined by using the GEOS-Chem chemical transport model version 12.2.1 (Bey et al., 2001; http://geos-chem.org, last access on 14 February 2020). GEOS-Chem is a global 3D chemical transport model capable of simulating global trace gas (more than 100 tracers) and aerosol distributions. The GEOS-Chem model implements a universal tropospheric-stratospheric Chemistry (UCX) mechanism (Eastham et al., 2014). All simulations were performed in a standard GEOS-Chem full-chemistry mode with a horizontal resolution of $2° \times 2.5°$ and were initialised for one year (July 2014 to July 2015) to remove the influence of the initial conditions. The model is driven by the Goddard Earth Observing System-Forward Processing (GEOS-FP) meteorological fields at a horizontal resolution of $2° \times 2.5°$ degraded from their native resolution of $0.25° \times 0.3125°$. The temporal resolutions are 1 hour (hr) for surface variables and boundary layer height, and 3 hr for other variables. The photolysis rates were obtained from the FAST-JX v7.0 photolysis scheme (Bian and Prather, 2002). Dry deposition was calculated by the resistance-in-series algorithm (Wesely, 1989; Zhang et al., 2001), and wet deposition followed that of Liu et al. (2001). The GEOS-Chem model outputs 72 vertical layers of HCHO VMR concentration ranging from the surface to 0.01 hPa at a temporal resolution of 1 hr. This study only considered the HCHO simulations from 2015 to 2019 in the grid box containing Hefei (31.52°–32.11°N by 116.53°–118.02°E).

Emissions in GEOS-Chem are processed through the Harvard–NASA Emission Component (HEMCO) (Keller et al., 2014). The anthropogenic emissions were from the Community Emissions

Data System (CEDS; the latest 2015 condition is used for the model simulation) inventory (Hoesly et al., 2018), which overwrites regional emissions over Asia with the MIX inventory (Li et al., 2017; Lu et al., 2019; Zheng et al., 2018). In particular, the latest Chinese anthropogenic emissions for 2016 and 2017 from the Multi-resolution Emission Inventory for China (MEIC; http://www.meicmodel.org, last access: 14 April 2020) were implemented (Lu et al., 2019; Zheng et al., 2018). The MEIC is a bottom-up emission inventory with particular improvements in the accuracy of unit-based power plant emission estimates (Liu et al., 2015), vehicle emission modelling (Zheng et al., 2014), and the NMVOCs speciation method (Li et al., 2014). Global biomass burning emissions were from the Global Fire Emissions Database version 4 (GFED4) inventory (Giglio et al., 2013). Biogenic emissions were from the Model of Emissions of Gases and Aerosols from Nature (MEGAN version 2.1) inventory (Guenther et al., 2012), and soil $NO_x$ emissions were calculated following the method of Hudman et al. (2010, 2012). Mixing ratios of $CH_4$ are prescribed in the model based on spatially interpolated monthly mean surface $CH_4$ observations from the NOAA Global Monitoring Division for 1983–2016 and are extended to 2020 using the linear extrapolation of local 2011–2016 trends (Murray, 2016).

Total emissions of all atmospheric compounds in 2016 and 2017 over China by category are summarized in Table 2. In this study, we separated the anthropogenic emissions into fossil fuel and biofuel emissions. The global biofuel inventory is only available for the year 2015. The number of atmospheric compounds and the emission amounts in the biofuel emission inventory are much smaller than those in fossil fuel emission inventory. In addition, the combination of biogenic and biomass burning emissions is referred to as natural emission. Total annual Chinese anthropogenic emissions of $NO_x$ and NMVOCs are, respectively, 22.5 and 28.4 Tg in 2016 and 22.0 and 28.6 Tg in 2017. Total annual Chinese natural emissions of $NO_x$ and NMVOCs are, respectively, 1.74 and 27.16 Tg in 2016 and 1.56 and 28.02 Tg in 2017. The anthropogenic emissions of all atmospheric compounds are dominated by fossil fuel emissions and the natural NMVOCs emissions are dominated by biogenic emissions. We cannot separate the $CH_4$ emissions into anthropogenic and natural emissions since the $CH_4$ concentrations are prescribed based on NOAA measurements, and hence cannot be shut off the same way as for other emission inventories. We find 1% increases in $CH_4$ concentration over eastern China in 2017 relative to 2016 (Lu et al., 2019).

**2.5.2 GEOS-Chem model configurations**

First, we conducted a standard full chemistry simulation (hereafter BASE) including all emission inventories as described in Table 2 and took it as the reference. Then, we conducted a series of sensitivity simulations to assess the change of each sensitivity simulation relative to the BASE simulation. We followed the method of Franco et al. (2016) and did not shut off the $CH_4$ inventory in all sensitivity simulations, i.e., $CH_4$ concentrations were still derived from the NOAA measurements as for the BASE simulation. The model configurations used in this study are summarised in Table 3 and were designed as follows.

(i) To analyse the contributions of different emission categories, each individual emission inventory was shut off to evaluate the change of the simulation in the presence of all other emission inventories. Thus, the relative contribution of each emission category was estimated as the relative difference between the GEOS-Chem simulation in the presence and absence of that emission inventory. We have conducted four such sensitivity simulations by shutting off (1) fossil fuel emission inventory (including emissions from agriculture, industry, power plant, residential, and

transport), (2) biogenic emission inventory, (3) biomass burning emission inventory, and (4) biofuel emission inventory (Table 2). When an emission inventory was shut off, global emissions of all atmospheric compounds in this inventory were set to be zero.

(ii) To analyse the contributions of different geographical regions, the emission inventory clusters within each geographical region were shut off to assess the change of the simulation in the presence of emissions outside that geographical region. Thus, the relative contribution of each geographical region was estimated as the relative difference between the GEOS-Chem simulation in the presence and absence of the emission inventory clusters within that geographical region. We have conducted five such sensitivity simulations by shutting off the emission inventory clusters within five geographical regions. Here the emission inventory clusters are defined as all emission inventories except $CH_4$ inventory in Table 2. When the emission inventory clusters in a specific region were shut off, emissions of all relevant atmospheric compounds within that region were set to be zero. The geographical regions are shown in Fig. 1(a) and the resulting delimitations are summarised in Table 3. The delimitations of these geographical regions are based on the levels of urbanization and industrialization in China. Region ① in Fig. 1(a) only covers a few sparsely city clusters representing the region with least population and industrialization in China (Lu et al. 2019). Regions ②, ④, and ⑤ cover the North China Plain (NCP), YRD, and Pearl River Delta (PRD) city clusters, respectively, which are the three most developed city clusters facing severe air pollution in China. Region ③ covers the Sichuan Basin (SCB) and central Yangtze River (CYR) city clusters with newly emerging severe air pollution in China.

Regional air quality is not only influenced by local emission but also by long range transport. In addition, a reduction in one pollutant may affect the conditions of many atmospheric compounds via a chain of complex atmospheric chemical reactions. Sensitivity simulations in this study were performed by shutting off all atmospheric compounds simultaneously rather than the HCHO precursors only. This approach provides an evaluation for the consequence of the recent clean air policies which affect not only HCHO precursors but also many other atmospheric pollutants (Zheng et al., 2018).

**3 FTIR HCHO dataset over Hefei**

The FTIR measurements taken with a solar intensity variation (SIV) of larger than 10% or retrievals with DOFS of less than 0.7 or root-mean-square (RMS) of fitting residuals of larger than 2% were excluded in this study. This filter criterion excluded the measurements seriously affected by instable weather conditions or by the *a priori* profile due to low measurement information content in less favourable observational conditions, e.g., around noontime when the probed atmosphere is thinner, or in winter when HCHO is less abundant. With this criterion, 12.4% of FTIR measurements were excluded in subsequent study. For the ground level in situ datasets provided by the CNEMC site, LP-DOAS and CRDS analysers, the measurements collected during maintenance, adjustments, and calibrations were excluded, as well as measurements collected during electricity failures.

**3.1 Comparison with LP-DOAS dataset**

The LP-DOAS ground level HCHO measurements nearest to each individual FTIR $X_{HCHO}$ measurement were included for comparison. The temporal difference between FTIR and LP-DOAS dataset is within ± 5 minutes. Correlation plots of FTIR $X_{HCHO}$ measurements against LP-DOAS ground level HCHO measurements are shown in Fig.3. The results show that the HCHO variability

observed by FTIR and LP-DOAS are in good agreement with a correlation coefficient ($r^2$) of 0.88. The amplitude of the LP-DOAS ground level measurements is on average 7.89 times that of the FTIR column-averaged measurements. This means HCHO column measurements at Hefei can be used as representative of near-surface conditions. As a result, this study used a constant factor of 7.89 to scale the column-averaged HCHO concentration to ground level HCHO concentration, or vice versa.

Over polluted atmosphere, the HCHO column measurements can be used as representative of near-surface conditions because HCHO is a tropospheric gas and has a vertical distribution that is heavily weighted toward the lower troposphere (Martin et al., 2004). As shown in Fig.2(c), the HCHO concentration decreased by 72.7% with an increase in the height from surface to 3 km and continued to decrease slowly in the troposphere above 3 km. The HCHO partial column below 3 km accounted for 67.1% of HCHO total column. This percentage is expected to show less seasonal variation since the shape of the retrieved profile is very similar to the shape of the *a priori* profile due to the low DOFS (Fig. 2 (c)). Many studies have taken advantage of this favorable vertical distribution of HCHO to derive surface emissions of VOCs from space (e.g. Palmer et al., 2003; Millet et al., 2008; Boersma et al., 2009; Stavrakou et al., 2009; Fortems-Cheiney et al., 2012; Barkley et al., 2013; Marais et al., 2014; Streets et al., 2013; Gao et al., 2018). Meanwhile, the use of HCHO column measurements to explore tropospheric $O_3$ sensitivities has been the subject of several past studies, which disclosed that this diagnosis of $O_3$ production rate ($PO_3$) is consistent with the findings of surface photochemistry (eg., Martin et al., 2004; Duncan et al., 2010; Choi et al., 2012; Witte et al., 2011; Jin and Holloway, 2015; Mahajan et al., 2015; Jin et al., 2017; Schroeder et al. 2017). Source separation of atmospheric HCHO in Hong et al. (2018) and Su et al. (2019) also taken the advantage that column measurements of HCHO are fairly representative of near-surface conditions.

**3.2 Seasonal and interannual variabilities**

We have used the bootstrap resampling method of Gardiner et al. (2008) with a 3[rd] Fourier series plus a linear function to fit FTIR daily mean time series of $X_{HCHO}$ (Fig.4(a)). Generally, the measured features in terms of seasonality and interannual variability from 2015 to 2019 can be reproduced by the bootstrap resampling model with a correlation coefficient ($r^2$) of 0.81. The FTIR $X_{HCHO}$ roughly increases over time for the first half of the year and decreases over time for the second half of the year (Fig. 4(b)). The $X_{HCHO}$ reached a maximum monthly mean value of ($1.1 \pm 0.27$) ppbv in July and a minimum monthly mean value of ($0.4 \pm 0.11$) ppbv in January. The FTIR $X_{HCHO}$ values in July were on average 1.75 times higher than those in January. In term of HCHO total column, the maximum and minimum monthly mean values are ($1.68 \pm 0.39$) and ($0.66 \pm 0.16$) $\times 10^{16}$ molec cm$^{-2}$, respectively. The annual mean values of $X_{HCHO}$ and HCHO total column over Hefei are ($0.55 \pm 0.14$) ppbv and ($1.04 \pm 0.27$) $\times 10^{16}$ molec cm$^{-2}$, respectively. As commonly observed, the seasonal HCHO enhancements spanned a wide range of -50.0% to 60.0% depending on the season and measurement time (Fig. 4 (b)). The observed HCHO time series from 2015 to 2019 showed a positive change rate of ($2.38 \pm 0.71$) % per year (Fig. 4 (a)). This positive change rate in HCHO concentration over China was in agreement with the positive trends observed by the spaceborne Ozone Monitoring Instrument (OMI) from 2004 to 2014 by De Smedt et al. (2015) and from 2005 to 2017 by Zhang et al. (2019).

Recently, Vigouroux et al. (2018) presented an unprecedented harmonized HCHO total column

dataset from 21 ground-based FTIR stations around the globe. These FTIR stations sample a wide
range of HCHO total columns from 0.1 to 2.2 $\times$ $10^{16}$ molec cm$^{-2}$ and are classified as clean,
intermediate, and high-level HCHO stations. Vigouroux et al. (2018) found that high levels of
HCHO are typically observed at the places which are affected by large anthropogenic emissions
such as Toronto and Mexico City (means of 0.95 and 2.21$\times10^{16}$ molec cm$^{-2}$), or affected by large
biogenic emissions such as Wollongong (mean of 0.79$\times10^{16}$ molec cm$^{-2}$) and Porto Velho, located
at the edge of the Amazon rainforest (mean of 1.9$\times10^{16}$ molec cm$^{-2}$). In comparison, the Hefei site
is affected by both anthropogenic and biogenic emissions due to the surrounding megacity, wetlands
and cultivated lands (see section 2.1). The HCHO total columns at Hefei are comparable with those
at Toronto and are lower than those at Mexico City and Porto Velho. With the classification criteria
in Vigouroux et al. (2018), the Hefei site can be classified as a high-level HCHO station and has the
third highest levels of HCHO concentration around the globe.
**3.3 Separation between emitted and photochemical sources**
The CNEMC ground level CO and $O_x$ measurements nearest to each individual FTIR $X_{HCHO}$
measurement were included for source separation. The temporal difference between FTIR and
CNEMC dataset is within $\pm$ 30 minutes. For the polluted atmosphere over Hefei, it is impossible to
directly measure the background HCHO concentration and thus an empirical value derived previous
studies in the YRD region was used. According to the ground level measurements of HCHO at a
rural site in the YRD region by Ma et al. (2016) and Wang et al. (2015), the background level of
HCHO near the surface was approximately 1.0 ppbv in springtime. We scaled this background level
(1.0 ppbv) into column-averaged concentration with the scale factor deduced in section 3.1, and
coupled the resulting value with a 3$^{rd}$ Fourier series to reconcile the seasonal difference in HCHO
background. As a result, the fitting process in this study was initiated by assigning the background
with a 3$^{rd}$ Fourier series with an amplitude of 0.22 ppbv. Garcia et al. (2006) carried out a series of
sensitivity tests by using a series of empirical background concentrations to initialize the regression
analysis. Garcia et al. (2006) found that the percent fraction of emitted HCHO is almost constant in
all sensitivity tests, but the percent fractions of background and photochemical HCHO contributions
are anti-correlated, and scale linearly with the background value. The fact that photochemical
HCHO decreases as the background HCHO increases, suggests a relation of the background with
photochemistry rather than emission sources (Garcia et al., 2006). It is worth noting that
imperfections in source separation with this regression model are likely to become significant in
certain cases. In this study, photochemical HCHO production from CH$_4$ oxidation in the free
troposphere which can hardly be accounted for by the in situ tracers is in fact erroneously (or at least
partly) interpreted background HCHO. In addition, the measurements with large temporal variations
of HCHO/CO or HCHO/$O_x$ ratios generally can't be reproduced by this regression model. A more
sophisticated multi-regression model might be able to reduce the uncertainties, but this is beyond
the scope of present work.
Seasonal variabilities of absolute and relative contributions of emitted, photochemical, and
background sources to the observed $X_{HCHO}$ are shown in Fig. 5. The correlation coefficient value ($r^2$)
from the regression analysis indicates the proportion of HCHO measurements that can be
reproduced by the regression model (Green, 1998). The results indicate that this proportion is for
all subsets of dataset well above 80%, and up to 92%, reflecting that the CO-$O_x$ tracer pair – while
not perfect – generally replicates well the observations. Statistical modelling results for relative

contributions of different sources to the observed $X_{HCHO}$ from 2015 to 2019 are listed in Table 4. The relative contributions of emitted and photochemical sources spanned a wide range of values throughout the year; however, the relative contributions of the background source were roughly a constant value. Depending on measurement time and season, the relative contributions of emitted sources varied from 14.0% to 58.0%, and relative contributions of photochemical sources varied from 20% to 82%. On average, the relative contributions of emitted, photochemical, and background sources to the observed $X_{HCHO}$ from 2015 to 2019 were 29.0 ± 19.2%, 49.2 ± 18.5%, and 21.8 ± 6.1%, respectively. As evidenced in Table 2, the emitted HCHO are mainly from fossil fuel and biomass burning emissions. In addition to oxidation of $CH_4$, oxidations of both fossil fuel and biogenic NMVOCs could have large contributions to photochemical HCHO, which will be discussed in detail in section 4.2.

All measurements were further separated into emitted-dominated or photochemical-dominated measurements according to a larger contribution to the observed $X_{HCHO}$ (Table 4). Generally, photochemical HCHO is the dominant source of atmospheric HCHO over Hefei for most of the year (68.1%). The largest contrast between photochemical and emitted in terms of domination percent fraction occurs in the afternoon (after 12:00 a.m. local time (LT)) in summer and autumn (JJA/SON) season when the photochemistry for HCHO formation is enhanced. Indeed, the LP-DOAS measurements in this study and many previous studies with either in situ dataset (Li et al., 2010, Lui et al., 2017; Ma et al., 2016, Wang et al., 2015) or remote sensing dataset (De Smedt et al., 2015; Vigouroux et al., 2018; Franco et al., 2016; Peters et al., 2012) disclosed that the typical diurnal modulation of HCHO at mid-latitudes shows a pronounced peak in the early afternoon.

## 3.4 Hydroxyl (OH) radical production from HCHO

Photolysis plays a significant role in the degradation of HCHO and one of its two photo dissociative paths provides a large source of OH radicals. The photolysis pathways of HCHO to form the OH radical are summarised as follows (Chapter 6, Seinfeld and Pandis, 2016):

$$HCHO + h\nu \rightarrow H + HCO \ (\lambda \leq 370\,\text{nm}) \rightarrow H_2 + CO \qquad (9)$$

$$H + O_2 \rightarrow HO_2 \qquad (10)$$

$$HCO + O_2 \rightarrow HO_2 + CO \qquad (11)$$

$$HO_2 + NO \rightarrow OH + NO_2 \qquad (12)$$

In air, the photolysis of HCHO first generates a hydroperoxyl ($HO_2$) radical at wavelengths below 370 nm. Then, $HO_2$ rapidly reacts with NO to generate the OH radical, and subsequently affects the oxidative capacity of the atmosphere (Possanzini et al., 2002; Volkamer et al., 2010). Under steady-state conditions, the total OH radical production rate from the photolysis of HCHO through the above chain of reactions is:

$$\frac{p[OH]_{HCHO}}{dt} = 2J_a[HCHO] \qquad (13)$$

where [HCHO] is the concentration of HCHO and $J_a$ is the photolysis constant of reaction (9). In comparison, applying steady state to reactions (14) – (16),

$$O_3 + h\nu \rightarrow O_2 + O(1D) \qquad (14)$$

$$O(1D) + M_{air} \rightarrow O(3P) + M_{air} \qquad (15)$$

$$O(1D) + H_2O \rightarrow 2OH \qquad (16)$$

the total OH radical production rate from $O_3$ is given by (Chapter 6, Seinfeld and Pandis, 2016),

$$\frac{p[OH]_{O_3}}{dt} = 2J_c \frac{k_e}{k_d} \frac{[O_3][H_2O]}{[M_{air}]} \tag{17}$$

where $[O_3]$, $[H_2O]$, and $[M_{air}]$ are the concentrations of $O_3$, $H_2O$, and air, respectively; $J_c$ is the photolysis constant of reaction (14); and $k_d$ and $k_e$ are the reaction rate coefficients for (15) and (16), respectively.

In this study, photolysis rate constants for HCHO and $O_3$ were available from the GEOS-Chem simulation, and the reaction rate coefficients were calculated according to a well-known procedure (Table B1; Seinfeld and Pandis, 2016). Surface $H_2O$ concentrations were available from an in situ CRDS analyser. For the atmosphere $N_2/O_2$ mixture at 298 K, the values of $k_d$ and $k_e$ are $2.9 \times 10^{-11}$ and $2.2 \times 10^{-10}$ cm$^3$ molecule$^{-1}$s$^{-1}$, respectively. The air concentration $[M_{air}]$ is 0.99 molecules cm$^{-3}$ (Chapter 6, Seinfeld and Pandis, 2016). The concentrations of HCHO and $O_3$ were based on FTIR observations and the CNEMC network, respectively. To reconcile the difference between the ground level concentration and column-averaged concentration, all individual FTIR $X_{HCHO}$ concentrations were converted to ground level VMRs with the scale factor deduced in section 3.1. For the ground level $H_2O$ and $O_3$ datasets, only measurements nearest to each individual FTIR measurement were considered. The temporal difference between FTIR and CNEMC (CRDS) is within $\pm$ 30 minute ($\pm$ 30 second).

The total OH radical production rates from the photolysis of HCHO and $O_3$ from 2015 to 2019 over Hefei calculated via equations (13) and (17) are shown in Fig. 6. For both gases, the OH radical production rates in summertime are higher than those in wintertime. Generally, OH radical production rates from the photolysis of HCHO are comparable with those from the photolysis of $O_3$ in all seasons. In wintertime when the concentrations in $O_3$ and $H_2O$ are low, or when emitted sources dominate the HCHO measurements, OH radical production rates from HCHO photolysis are higher than those from $O_3$ photolysis. In other seasons, when the concentrations in $O_3$ and $H_2O$ are high, or when photochemical sources dominate the HCHO measurements, OH radical production rates from HCHO photolysis are lower than those from $O_3$ photolysis. On average, the OH production rate from $O_3$ photolysis is 6.1% higher than that from HCHO photolysis. The results clearly indicate that HCHO photolysis was by far an important source of OH radical over eastern China during all sunlight hours of both summer and winter days.

**4 Source attribution by GEOS-Chem sensitivity simulations**
**4.1 Model evaluation**

The GEOS-Chem model was used to evaluate relative contributions of various emission categories and geographical regions to the observed HCHO summertime enhancements. For model evaluation, the observed $X_{HCHO}$ seasonal cycle was compared to the GEOS-Chem BASE simulations to investigate the chemical model performance for the specifics of polluted regions over eastern China. As the vertical resolution of GEOS-Chem is different from the FTIR measurement, a smoothing correction was applied to the GEOS-Chem profiles. First, the GEOS-Chem daily mean profiles of HCHO were interpolated to the FTIR altitude grid to ensure a common altitude grid. Since the FTIR instrument only operates during daytime, the average for GEOS-Chem simulations is only performed during daytime from 9:00 to 17:00 LT. The interpolated profiles were then smoothed by the seasonal mean FTIR averaging kernels and *a priori* profiles (Rodgers, 2000; Rodgers and Connor, 2003). The GEOS-Chem $X_{HCHO}$ concentrations were calculated subsequently from the smoothed profiles by using the corresponding regridded air density profiles from the model.

Finally, the GEOS-Chem $X_{HCHO}$ time series only for the days with available FTIR observations were
averaged by month and compared with the FTIR monthly mean data.
Fig. 4 (a) shows the comparison of daily mean time series of $X_{HCHO}$ between the FTIR
observation and the smoothed GEOS-Chem model simulation from 2015 to 2019. Fig. 4 (b)
compares the seasonal cycles derived from Fig. 4 (a) for the days with available FTIR observations
only. The observed day-to-day variability cannot be always reproduced by the GEOS-Chem
simulation, especially in the trough and peak of the measurements (Fig. 4(a)). This can be partially
explained by the fact that many oxidation pathways of VOCs precursors leading the HCHO
production, which are numerous, might not be optimally implemented (especially very short-lived
VOCs) or merely not considered in the model (Franco et al., 2016). In addition, large uncertainties
remain concerning the various sources of precursor emissions, their geographical distributions and
how these sources can influence the air masses over polluted sites such as Hefei. Finally, GEOS-
Chem averages HCHO concentration over a large coverage area due to its relatively coarse spatial
resolution (here $2° \times 2.5°$). The Hefei site is located in a densely populated and industrialised area
in eastern China. The regional differences in HCHO concentration could aggravate the
inhomogeneity within the selected GEOS-Chem coverage grid cell, which also affects the
comparison with observations. Nevertheless, the measured feature in term of seasonal cycle of
HCHO loadings over Hefei can be reproduced by GEOS-Chem simulations with a correlation
coefficient ($r^2$) of 0.78 (Fig. 4(b)). The averaged difference between GEOS-Chem and FTIR dataset
(GEOS-Chem minus FTIR) is $-0.05 \pm 0.2$ ppbv ($-2.6 \pm 10.4\%$), which is within the FTIR uncertainty
budget. As a result, GEOS-Chem can simulate the concentration and seasonal variation of HCHO
for the heavily polluted regions over eastern China. Previous studies have also found that global
chemistry transport models were able to reproduce the absolute values as well as seasonal cycles of
the ground-based FTIR HCHO observations in the other parts of the world (Franco et al., 2016;
Vigouroux et al., 2018).
**4.2 Emission category contribution to HCHO enhancement**
In this part of the study, the summertime HCHO model simulations are analysed to assess the
contribution of each emission category to the maximum seasonal enhancements throughout the year,.
Fig. 7 (a) shows daily mean $X_{HCHO}$ time series averaged in the summers of 2015 to 2019 over Hefei
simulated by GEOS-Chem, according to the BASE and sensitivity (i.e., noFF, noBVOC, noBB, and
noBIOF) simulations. Fig. 7 (b) presents relative contribution of each emission category calculated
as the relative difference between the BASE simulation and the corresponding sensitivity simulation
(in %).
As can be seen in Fig. 7 (a) and (b), shutting off emission sources of fossil fuel and biogenic
significantly impacts the simulated HCHO summertime loadings over Hefei, with the $X_{HCHO}$ derived
from either the noFF or noBOVC simulations reduced by $10 - 65\%$ relative to the BASE simulation.
However, shutting off biomass burning and biofuel emissions have almost no effect on the simulated
HCHO summertime loadings over Hefei, with the $X_{HCHO}$ derived from either the noBB or noBIOF
simulations reduced by less than 3% relative to the BASE simulation. In addition, the variations of
the influences of noFF and noBOVC are also much larger than those of noBB and noBIOF.
Modelled $X_{HCHO}$ summertime simulations from 2015 to 2019 were on average reduced by 0.18, 0.23,
0.01, and 0.01 ppbv in the absence of fossil fuel, biogenic, biomass burning, and biofuel emission
inventories, respectively, which contribute 24.98, 29.81, 1.0, and 0.95% to the HCHO summertime

enhancements (Fig.A1). The anthropogenic emissions accounted for 25.93% and the natural emissions accounted for 30.81% of the HCHO summertime enhancements. Contributions of fossil fuel and biogenic emissions are much larger than those of biomass burning and biofuel emissions because of larger NMVOCs emissions from fossil fuel and biogenic sources (Table 2).

The remaining contribution was calculated as the difference between the BASE simulation and the sum of all emission contributions as estimated from the sensitivity simulations, and was 0.29 ppbv (43.27%). This remaining contribution can be largely attributed to the global $CH_4$ emissions and the nonlinear interactional effects among different sources which were not captured by the sensitivity simulations. Indeed, shutting off some emission sources in the GEOS-Chem sensitivity simulations eventually resulted in slightly enhanced HCHO amounts (by 1−1.5 %) compared to the BASE simulation, as shown in Fig. 7(b) for the noBIOF simulation and, to a lesser extent, for the noBB simulation during later summer. In these particular cases, shutting off an emission inventory may induce significantly lower concentrations in many atmospheric compounds globally, some of which mainly react with OH. This would lead to higher OH concentrations available for the oxidation of HCHO precursors, and eventually enhances the HCHO production from other emission categories (Franco et al., 2016). However, it is difficult to quantify the nonlinear impact of each individual emission category, since the types of atmospheric compounds and their concentrations in each emission category are different. Especially when the emissions of NO are suppressed, the impacts become hard to assess, since this compound plays a key role in both HCHO formation (through the degradation of peroxy radicals) and destruction (by contributing to the regeneration of OH) (Franco et al., 2016). Investigating the nonlinear impact of each individual emission category would require additional work that is beyond the scope of the present work.

These above sensitivity tests suggest that the oxidations of both NMVOCs and $CH_4$ (not included in the emission perturbations here) dominate the HCHO production and are the main drivers of its summertime enhancements over Hefei. This is different from Franco et al. (2016), which found that HCHO summertime loadings over Jungfraujoch, Swiss land were dominated by the oxidation of $CH_4$, and the contribution of NMVOCs was rather limited. For the HCHO loadings over Jungfraujoch, it is most likely that a large part of the short-lived NMVOCs are already oxidized before being transported to this high altitude site (3580 m a.s.l.). Hence these NMVOCs compounds do not contribute directly to the HCHO loadings over Jungfraujoch, although their biogenic secondary products can be transported to the upper troposphere and contribute to the HCHO abundance there (Franco et al., 2016). However, the low altitude Hefei site (30 m a.s.l.) is surrounded by megacity, wetlands or cultivated lands (see section 2.1). A large amount of NMVOCs compounds originating from both anthropogenic and natural emissions contributed directly to the HCHO summertime loadings over Hefei, resulting in a much larger NMVOCs contribution than that over Jungfraujoch.

**4.3 Geographical region contribution to HCHO enhancement**

We present in this section contribution of each geographical region in China to the observed HCHO summertime enhancements. Geographical delimitations of these regions are summarised in Table 3. Fig 8 (a) shows daily mean $X_{HCHO}$ time series averaged in the summers of 2015 to 2019 over Hefei simulated by GEOS-Chem, according to the BASE and sensitivity (i.e., noER, noCR, noNR, noWR, and noSR) simulations. Fig 8 (b) shows relative contribution of each geographical region calculated as the relative difference between the BASE simulation and the corresponding

sensitivity simulation (in %).

2         We can see from Fig. 8 (a) and (b) that shutting off emission clusters in eastern China (noER)

dominantly impacts the simulated HCHO summertime loadings over Hefei, with the $X_{HCHO}$ derived
from noER simulations reduced by a wide range of $20 - 70\%$ relative to the BASE simulation.
Shutting off emission clusters in either central (noER), northern (noNR), or southern (noSR) China
occasionally reduce the simulated HCHO summertime loadings over Hefei by an intermediate
amplitude of $10 - 30\%$. However, shutting off emission clusters in western China (noWR) has
almost no effect on the simulated HCHO summertime loadings over Hefei, with the $X_{HCHO}$ derived
from noWR simulations reduced by less than 2% relative to the BASE simulation. Modelled $X_{HCHO}$
summertime simulations from 2015 to 2019 were on average reduced by 0.33, 0.06, 0.03, 0.01, and
0.03 ppbv in the absence of the emission clusters in eastern China, central China, northern China,
western China, and southern China, respectively, which correspond to contributions of 44.36%,
7.24%, 4.2%, 0.98%, and 4.59% to the HCHO summertime enhancements (Fig. A2). The remaining
contribution was calculated as the difference between the BASE simulation and the sum of all
geographical sensitivity simulations and was 0.27 ppbv (38.62%). This remaining contribution can
be largely attributed to global $CH_4$ emissions, NMVOCs emissions outside China and the nonlinear
interactional effects among the geographical sensitivity simulations. Indeed, shutting off regional
emission clusters in the GEOS-Chem geographical sensitivity tests investigated here eventually
resulted in slightly enhanced HCHO amounts (by $0.5 - 2$ %) produced by GEOS-Chem compared
to the BASE simulation, as shown in Fig. 8 (b) for the noSR simulations during later summer. It is
worth noting that the remaining contribution here is 4.65% lower than that in section 4.2 (without
global $CH_4$ emissions shut off in both cases), indicating that the nonlinear effects with emission
sources shut off globally are larger than those with regional emission clusters shut off.

24         As a short-lived species (a few hours), the primarily emitted HCHO is heavily contributed from

emissions at local and nearby regions. However, HCHO precursors originating from distant areas
can be transported to the Hefei site under favourable weather conditions, and thus contribute to
photochemical HCHO formation. In addition, atmospheric compounds, originating from sources
either nearby or in distant areas and affecting the chemistry of HCHO or its precursors, could
contribute to photochemical HCHO formation or background. As a result, in the vicinity of the
observation site, emissions over eastern China dominated both the emitted and photochemical
HCHO. Emissions outside eastern China mainly contributed to the photochemical or background
HCHO at the observation site because of long-distance transport. Indeed, the sensitivity tests suggest
that the NMVOCs related HCHO summertime enhancements were exclusively dominated by the
emissions within eastern China.

35         The emissions in western China are typically lower than those in other parts of China because

of lower population and industries in the region (Lu et al., 2019; Zheng et al., 2018). The strong
easterly and the south-westerly flows prevail in the lower troposphere during the summer Asian
monsoon, including the South Asian summer monsoon and East Asian summer monsoon (Liu et al.,
2003; Wu et al., 2012). Therefore, the western China has the lowest contribution to the observed
HCHO summertime enhancements due to the lowest HCHO precursor emissions and few air masses
transported from this region during the summer Asian monsoon.
**4.4 Potential factors drive interannual variability of HCHO**

43         In this study, we use previous HCHO measurements at a rural site in the YRD region to

represent the background HCHO concentration in the "clean" atmosphere over Hefei, and assume
its amplitude to be constant over years. As a result, the observed interannual variability of HCHO
from 2015 to 2019 was not driven by the background portion but by either emitted or photochemical
portions, or both. China has implemented a series of active clean air policies since 2013 to mitigate
severe air pollution problems ( Sun et al. 2020; Zhang et al. 2019; Zheng et al. 2018). Since then
the anthropogenic emissions of major air pollutants have decreased, and the overall air quality has
substantially improved (Sun et al. 2020; Zhang et al. 2019; Zheng et al. 2018). The Prevention and
Control of Atmospheric Pollution also included the prohibition of crop residue burning over China
in 2015 because crop residue burning emissions can result in poor air quality
(http://www.chinalaw.gov.cn, last access on 19 June 2020), leading to dramatical decrease in the
crop residue burning events over China since then (Sun et al. 2020). Indeed, as evidenced in Table
2, the anthropogenic and biomass burning emissions of many air pollutants, such as HCHO, sulphur
dioxide ($SO_2$), $NO_x$, TSP (particulate matter with an aerodynamic diameter of 100 μm or less),
particulate matter 2.5 ($PM_{2.5}$), particulate matter 10 ($PM_{10}$), CO, black carbon (BC), and organic
carbon (OC), showed decreases in 2017 relative to 2016 (Lu et al., 2019; Zhang et al. 2019; Zheng
et al. 2018).

17       Anthropogenic and biomass burning HCHO emissions showed relative change rates of -2.0%

and -17.0%, respectively, resulting in a total change rate of -9.5% in 2017 relative to 2016. As for
photochemical HCHO, biomass burning emissions of its NMVOCs precursors showed a significant
negative change rate of -17.0% in 2017 relative to 2016 as consequence of the prohibition of crop
residue burning over China. However, both anthropogenic and biogenic emissions of NMVOCs
showed positive change rates of 1.0% and 6.4%, respectively, in 2017 relative to 2016. When taken
all emission categories into account, NMVOCs emissions were increased by 1.9% in 2017 relative
to 2016. Furthermore, as an important precursor of HCHO, $CH_4$ emissions over eastern China were
increased by approximate 1% in 2017 relative to 2016 (Table 2). As a result, the observed increasing
change rate of HCHO from 2015 to 2019 can be, to a large extent, attributed to the increase in
photochemical HCHO resulting from increasing change rates of both NMVOCs and $CH_4$, which
overwhelmed the decrease in emitted HCHO.
**5 Concluding remarks**

30       China has implemented a series of active clean air policies in recent years to mitigate severe

air pollution problems. Therefore, the emissions of major air pollutants have decreased and the
overall air quality across China has substantially improved. However, the volatile organic
compounds (VOCs) emissions, which are key precursors of formaldehyde (HCHO) and ozone ($O_3$),
are still increasing because the current clean air policies in China lack mitigation measures for VOCs.

35       This study mapped the drivers of the observed variability in HCHO from 2015 to 2019 over

Hefei, eastern China using ground-based high-resolution Fourier transform infrared (FTIR)
spectroscopy and GEOS-Chem model simulations. The column-averaged dry air mole fractions of
HCHO ($X_{HCHO}$) reached a maximum monthly mean value of (1.1 ± 0.27) ppbv in July and a
minimum monthly mean value of (0.4 ± 0.11) ppbv in January. FTIR $X_{HCHO}$ concentrations in July
were on average 1.75 times higher than those in January. The $X_{HCHO}$ time series from 2015 to 2019
over Hefei showed a positive change rate of (2.38 ± 0.71) % per year. The variability of $X_{HCHO}$
observed by FTIR at Hefei is in good agreement with that of the ground level HCHO measurements
provided by a long path differential optical absorption spectroscopy (LP-DOAS) instrument and

thus the FTIR column measurements can be used as representative of near-surface conditions. The relative contributions of emitted and photochemical sources to the observed HCHO were analysed using ground level CO and $O_x$ ($O_3$ + $NO_2$) as tracers for emitted and photochemical HCHO, respectively. On average, the contributions of emitted, photochemical, and background sources to the observed $X_{HCHO}$ from 2015 to 2019 were 29.0 ± 19.2%, 49.2 ± 18.5%, and 21.8 ± 6.1%, respectively. The photochemical HCHO was the dominant source of atmospheric HCHO over Hefei for most of the year (68.1%). In the studied years, total hydroxyl (OH) radical production rates from the photolysis of HCHO and $O_3$ were comparable. The HCHO photolysis was by far an important source of OH radicals over Hefei during all sunlight hours of both summer and winter days.

We found the GEOS-Chem model can simulate the concentrations and seasonal variations of HCHO for the heavily polluted regions over eastern China and thus it can be used for source attribution. Contributions of different emission categories and geographical regions in China to the observed HCHO were determined by using a series of GEOS-Chem model sensitivity simulations. The oxidations of both $CH_4$ (methane) and nonmethane VOCs (NMVOCs) dominate the HCHO production over Hefei and constitute the main driver of its summertime enhancements. The NMVOCs and $CH_4$ emissions accounted for about 56.73% and 43.27% of the HCHO summertime enhancements over Hefei, respectively. The NMVOCs related HCHO summertime enhancements were exclusively dominated by the emissions within eastern China. The observed increasing change rate of HCHO from 2015 to 2019 over Hefei is attributed to the increase in photochemical HCHO resulting from increasing change rates of both NMVOCs and $CH_4$, which overwhelmed the decrease in emitted HCHO.

This study can provide an evaluation of recent VOCs emissions and regional photochemical capacity in China. In addition, understanding the sources of HCHO is a necessary step for tackling the problems of poor air quality in eastern China and mitigating the emissions of pollutants.

*Data availability.* The FTIR HCHO measurements and GEOS-Chem sensitivity simulations in this study are available on request.

*Author contributions.* YS conceived the concept and prepared the paper with input from all co-authors. HY carried out the GEOS-Chem sensitivity simulations. The rest authors contributed to this work by providing refined data or constructive comments.

*Competing interests.* The authors declare that they have no conflict of interest.

*Acknowledgements.* This work is jointly supported by the National High Technology Research and Development Program of China (No.2019YFC0214802, No.2017YFC0210002, 2018YFC0213201, 2019YFC0214702, and 2016YFC0200404), the National Science Foundation of China (No. 41575021, No. 51778596, No. 41977184, and No.41775025), the Major Projects of High Resolution Earth Observation Systems of National Science and Technology(05-Y30B01-9001-19/20-3), the Sino-German Mobility programme (M-0036), and Anhui Province Natural Science Foundation of China (No. 2008085QD180). The processing and post processing environment for SFIT4 are provided by National Center for Atmospheric Research (NCAR), Boulder, Colorado, USA. The NDACC network is acknowledged for supplying the SFIT software. The LINEFIT code is provided by Frank Hase, Karlsruhe Institute of Technology (KIT), Institute for Meteorology and Climate Research (IMK-ASF), Germany. We thank the senate of Bremen, Germany for support. We thank

the FTIR group at university of Wollongong, Australia for help in setting up and operating the FTIR spectrometer at Hefei. We thank the FTIR group at Royal Belgian Institute for Space Aeronomy (BIRA-IASB), Belgium for providing harmonized HCHO retrieval setup. We thank the GEOS-Chem team for the support and Tsinghua University, China for providing the latest MEIC inventory.

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

# Tables

Table 1. Error budget and degrees of freedom (DOFS) for signal of the HCHO retrieval at Hefei

| Items | Input values | Error |
|---|---|---|
| Temperature uncertainty | SD of NCEP | 0.61% |
| Retrieval parameters uncertainty | * | 0.03% |
| Interfering species uncertainty | SD of WACCM | 0.09% |
| Measurement noise | $1/SNR^2$ | 1.59% |
| Smoothing uncertainty | * | 0.83% |
| **Total random error** | / | 1.71% |
| Background curvature uncertainty | 1% | 0.35% |
| Optical path difference uncertainty | 1% | 0.07% |
| Field of view uncertainty | 1% | 0.08% |
| Solar zenith angle uncertainty | 1% | 0.37% |
| Phase uncertainty | 1% | 0.61% |
| ILS uncertainty | 1% | 0.23% |
| Line temperature broadening uncertainty | 10% | 0.49% |
| Line pressure broadening uncertainty | 10% | 6.60% |
| Line intensity uncertainty | 10% | 9.04% |
| **Total systematic error** | / | 11.24% |
| **Total error** | / | 12.29% |
| **DOFS (-)** | / | 1.09 |

* These input values used for error estimation are prescribed from the retrieval output

Table 2. Total emissions of all atmospheric compounds over China used in the GEOS-Chem model.

**Fossil fuel emissions (Tg)**

| Year | HCHO | $SO_2$ | $NO_x$ | NMVOCs | $NH_3$ | CO | TSP | $PM_{10}$ | $PM_{2.5}$ | BC | OC | $CO_2$ |
|---|---|---|---|---|---|---|---|---|---|---|---|---|
| 2016 | 0.31 | 13.4 | 22.5 | 28.4 | 10.3 | 141.9 | 17.9 | 10.8 | 8.1 | 1.3 | 2.3 | 10290.6 |
| 2017 | 0.30 | 10.5 | 22.0 | 28.6 | 10.3 | 136.2 | 16.7 | 10.2 | 7.6 | 1.3 | 2.1 | 10434.3 |
| Change | -2% | -22% | -2% | 1% | 0% | -4% | -7% | -6% | -6% | 0% | -9% | 1.4% |

**Biofuel emissions (Tg)**

| Year | HCHO | NMVOCs | CO | $NO_x$ |
|---|---|---|---|---|
| 2015 | 0.03 | 0.23 | 3.19 | 0.09 |

**Biomass burning emissions (Tg)**

| Year | HCHO | $SO_2$ | $NO_x$ | NMVOCs | $NH_3$ | CO | $PM_{2.5}$ | BC | OC | $CO_2$ |
|---|---|---|---|---|---|---|---|---|---|---|
| 2016 | 0.29 | 0.12 | 0.78 | 3.96 | 0.31 | 17.64 | 1.85 | 0.10 | 0.98 | 284.72 |
| 2017 | 0.24 | 0.09 | 0.64 | 3.32 | 0.24 | 14.00 | 1.41 | 0.08 | 0.75 | 229.68 |
| Change | -17% | -25% | -18% | -17% | -23% | -21% | -24% | -20% | -23% | -19% |

**Biogenic emissions (Tg)**

| Year | NMVOCs | $NO_x$ |
|---|---|---|
| 2016 | 23.2 | 0.96 |
| 2017 | 24.7 | 0.92 |
| Change | 6.4% | -4% |

**$CH_4$ emissions**

Extrapolation from the NOAA measurements. The relative change rate of $CH_4$ over eastern China in 2017 relative to 2016 is approximate 1%.

Table 3. GEOS-Chem model configurations and delimitations of all geographical regions. For all sensitivity
simulations, the $CH_4$ emission inventory is always switched on.

| Simulation | Region | Description |
|---|---|---|
| BASE | Global | Standard full chemistry simulation implemented all emission inventories at the same time. The BASE simulation is taken as the reference and used for model evaluation |
| noFF | Global | Same as BASE but without global fossil fuel emissions |
| noBVOC | Global | Same as BASE but without global biogenic emissions |
| noBB | Global | Same as BASE but without global biomass burning emissions |
| noBIOF | Global | Same as BASE but without global biofuel emissions |
| Rest1 | Global | Difference between BASE and the sum of FF, BVOC, BB, and BIOF contributions |
| noWR | 78.6° E – 103.4° E; 27.6°N - 48.8°N | Same as BASE but without anthropogenic and natural emissions within western China (WR), i.e., region ① in Fig. 1(a) |
| noNR | 103.4°E – 129.8°E; 34.6°N – 53.5°N | Same as BASE but without anthropogenic and natural emissions within northern China (NR), i.e., region ② in Fig. 1(a) |
| noCR | 103.4°E – 115.6°E; 27.6°N – 34.6°N | Same as BASE but without anthropogenic and natural emissions within central China (CR), i.e., region ③ in Fig. 1(a) |
| noER | 115.6°E – 123.6°E; 21.0°N – 34.6°N | Same as BASE but without anthropogenic and natural emissions within eastern China (ER), i.e., region ④ in Fig. 1(a) |
| noSR | 98.1°E – 115.6°E; 21.0°N – 27.6°N | Same as BASE but without anthropogenic and natural emissions within southern China (SR), i.e., region ⑤ in Fig. 1(a) |
| Rest2 | Rest of world | Difference between BASE and the sum of WR, NR, CR, ER, and SR contributions |

Table 4. Statistical modelling results for relative contributions of different sources to the observed $X_{HCHO}$ from
2015 to 2019 over Hefei, eastern China

| Items | Total N (%) | Emission domination N (%) | Photochemical domination N (%) | Background domination N (%) |
|---|---|---|---|---|
| All | 1502 (100%) | 480 (31.9%)[†] | 1022 (68.1%) | 0 (0) |
| Before 12:00 (LT) | 727 (48.4%) | 322 (21.4%) | 405 (27.0%) | 0 (0) |
| After 12:00 (LT) | 775 (51.6%) | 158 (10.5%) | 617 (41.1%) | 0 (0) |
| JJA/SON | 890 (59.3%) | 287 (19.1%) | 603 (40.1%) | 0 (0) |
| DJF/MAM | 612 (40.7%) | 193 (12.8%) | 419 (27.9%) | 0 (0) |
| $d\% > 0\%$[††] | 717 (47.7%) | 273 (18.2%) | 444 (29.6%) | 0 (0) |
| $d\% < 0\%$ | 785 (52.3%) | 207 (13.8%) | 578 (38.5%) | 0 (0) |
| Contribution | 100% | $29.0 \pm 19.2\%$[†††] | $49.2 \pm 18.5\%$ | $21.8 \pm 6.1\%$ |

[†]There are 480 measurements dominated by emitted source which accounts for 31.9% of all measurements
[††]Larger than the seasonal mean value (see equation (3) for detail).
[†††]The mean contribution of emitted source is $29.0 \pm 19.2\%$.

# Figures

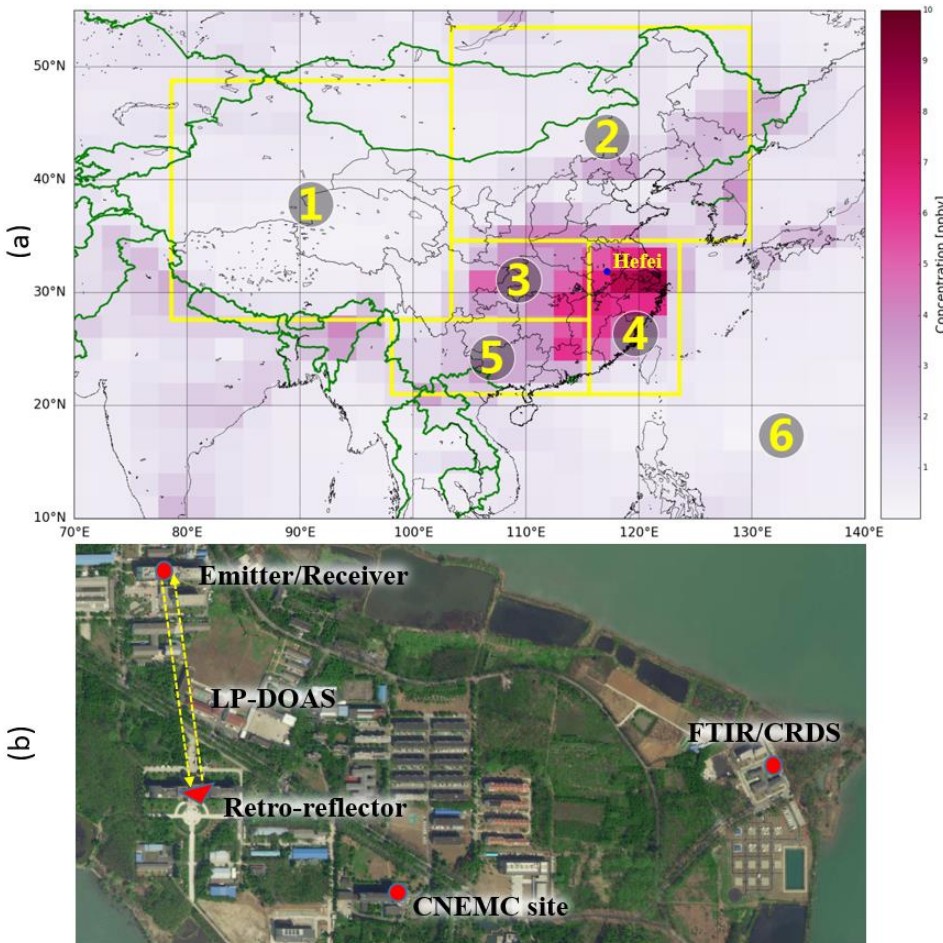

Fig. 1. (a) Location of the FTIR site at Hefei, eastern China and geographical regions used in GEOS-Chem sensitivity simulation. See Table 3 for latitude and longitude delimitation of each region. GEOS-Chem HCHO simulations on 24 July 2016 were selected for demonstration of summertime enhancement over eastern China. Region ① covers few sparsely city clusters representing the region with least population and industrialization in China. Regions ②, ④, and ⑤ cover the North China Plain (NCP), Yangtze River Delta (YRD), and Pearl River Delta (PRD) city clusters, respectively, which are the three most developed city clusters with severe air pollution in China. Region ③ covers the Sichuan Basin (SCB) and central Yangtze River (CYR) city clusters with newly emerging severe air pollution in China. (b) An overview of the location of the FTIR site, the CNEMC site and the optical path of the xenon LP-DOAS instrument.

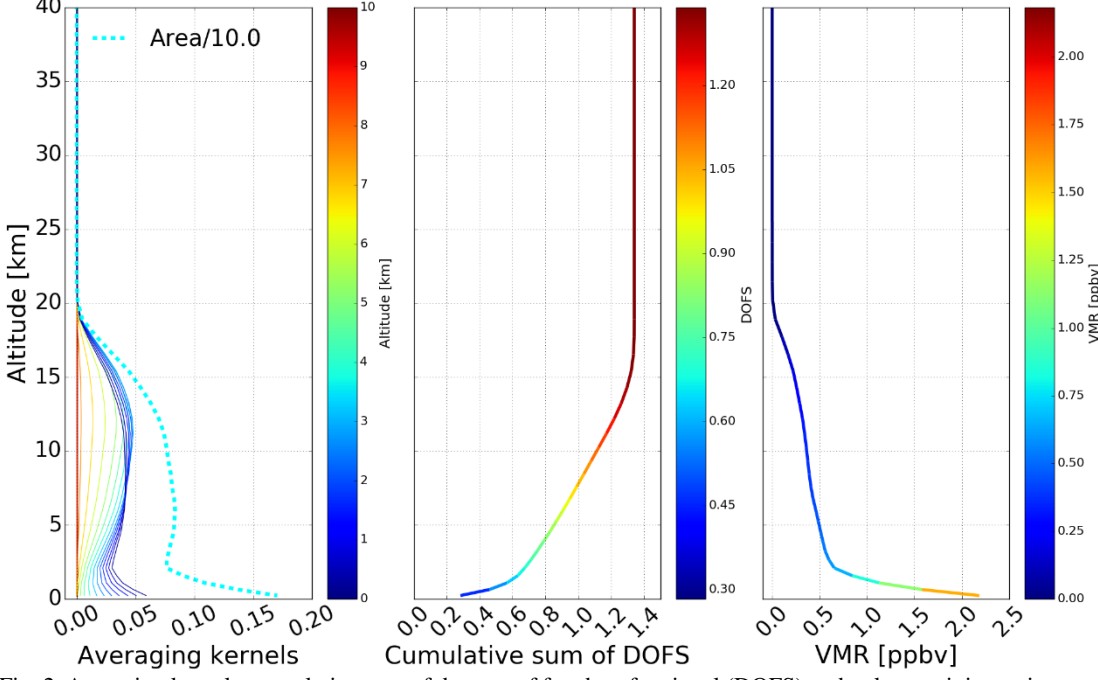

Fig. 2. Averaging kernels, cumulative sum of degrees of freedom for signal (DOFS) and volume mixing ratio (VMR) profile of randomly selected HCHO retrievals at Hefei, eastern China.

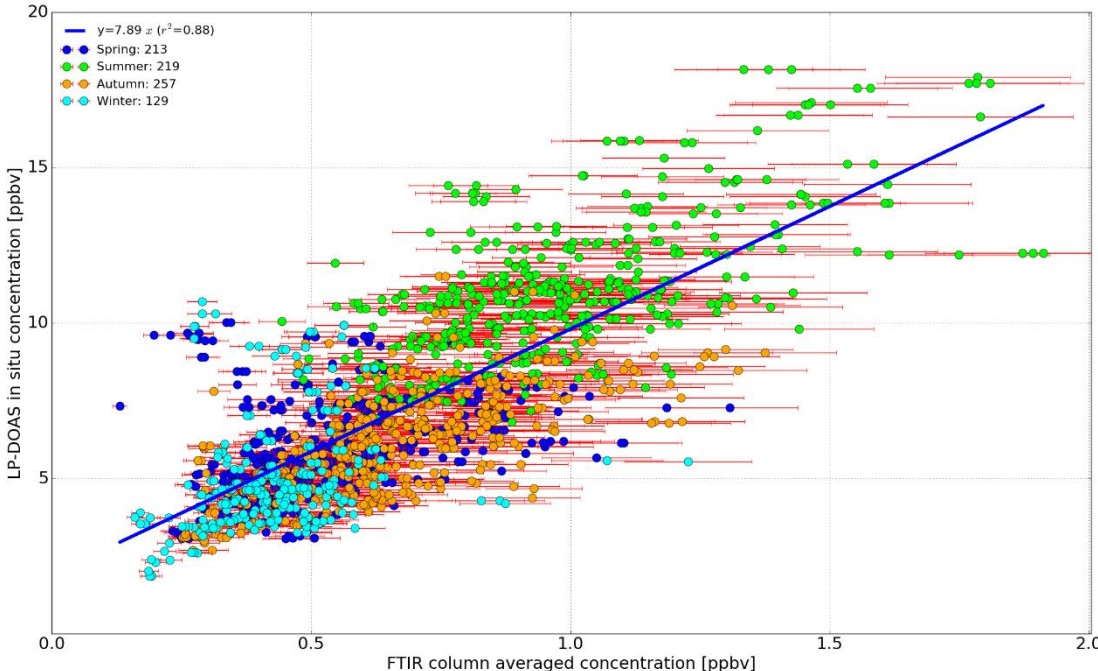

Fig. 3. Correlation plots of FTIR $X_{HCHO}$ measurements against LP-DOAS ground level HCHO measurements. The blue lines are linear fitted curves of respective scatter points. All concurrent data pairs were grouped by season. The number of data pairs within each season was also included.

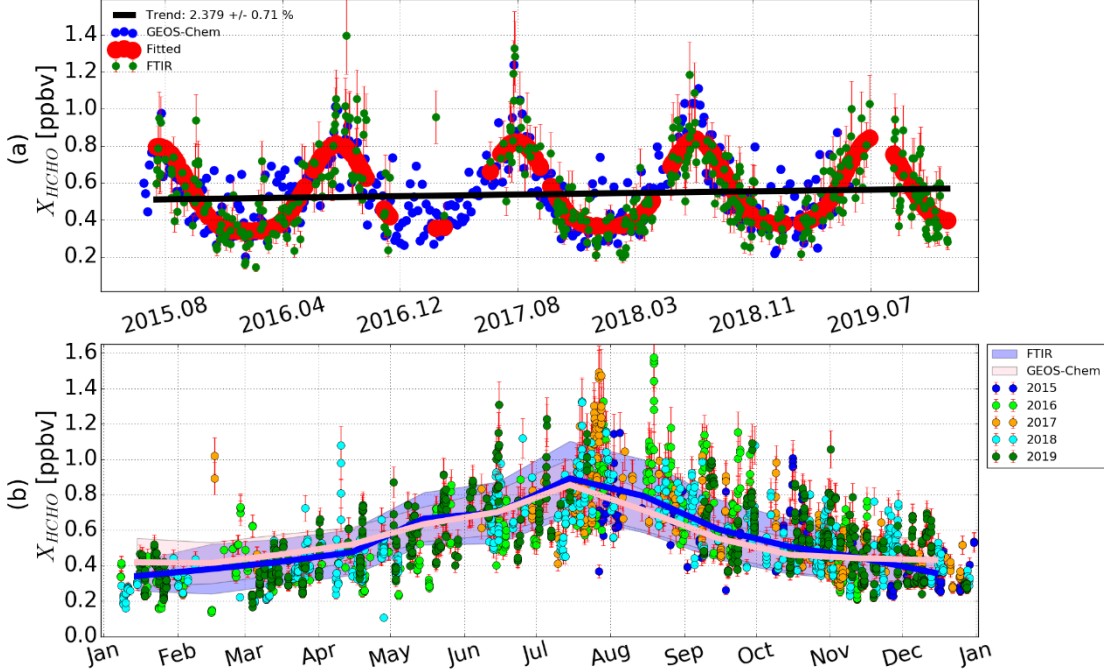

Fig. 4. Daily mean time series of $X_{HCHO}$ comparison between FTIR observation and GEOS-Chem model BASE
simulation from 2015 to 2019 over Hefei, China (a). The seasonality and interannual change rate are represented by
red dots and black line, respectively, which are fitted by using a bootstrap resampling model with a 3$^{rd}$ Fourier series
plus a linear function. (b) Seasonal variations of $X_{HCHO}$ by FTIR and GEOS-Chem simulation. Bold curves and the
shadows are monthly mean values and the 1-σ standard variations, respectively. Vertical error bars represent retrieval
uncertainties.

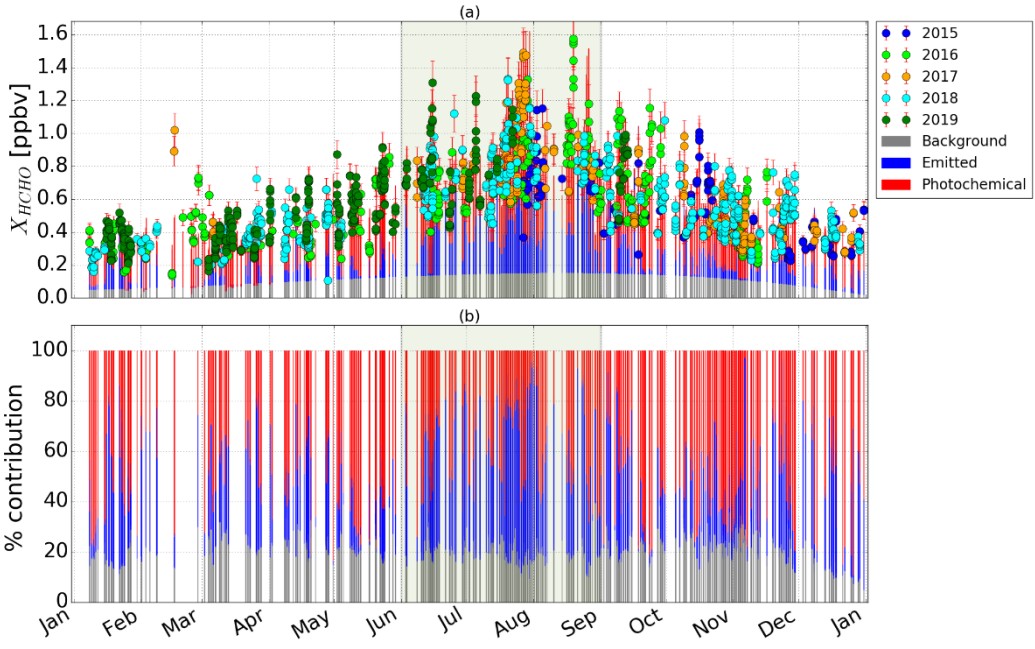

Fig. 5. (a) Separation between emitted and photochemical HCHO by using CO - O$_x$ tracers to model FTIR
observations. (b) Relative contributions of emitted, photochemical and background sources to the observed $X_{HCHO}$
from 2015 to 2019 over Hefei, eastern China. The grey vertical shaded area indicates summertime measurements.

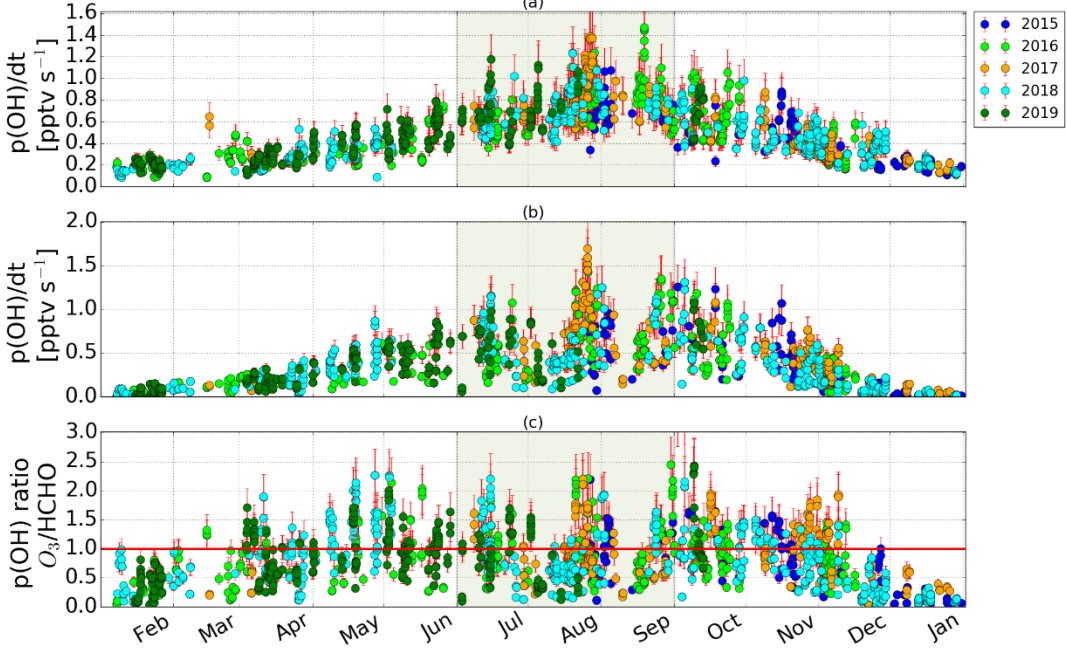

Fig. 6. Total OH radical production rates from the photolysis of HCHO (a) and O₃ (b) from 2015 to 2019 over Hefei,
eastern China. (c) The ratios of OH radical production rates from O₃ photolysis to that from HCHO photolysis. The
grey vertical shaded area indicates summertime measurements. The red line denotes one-to-one line.

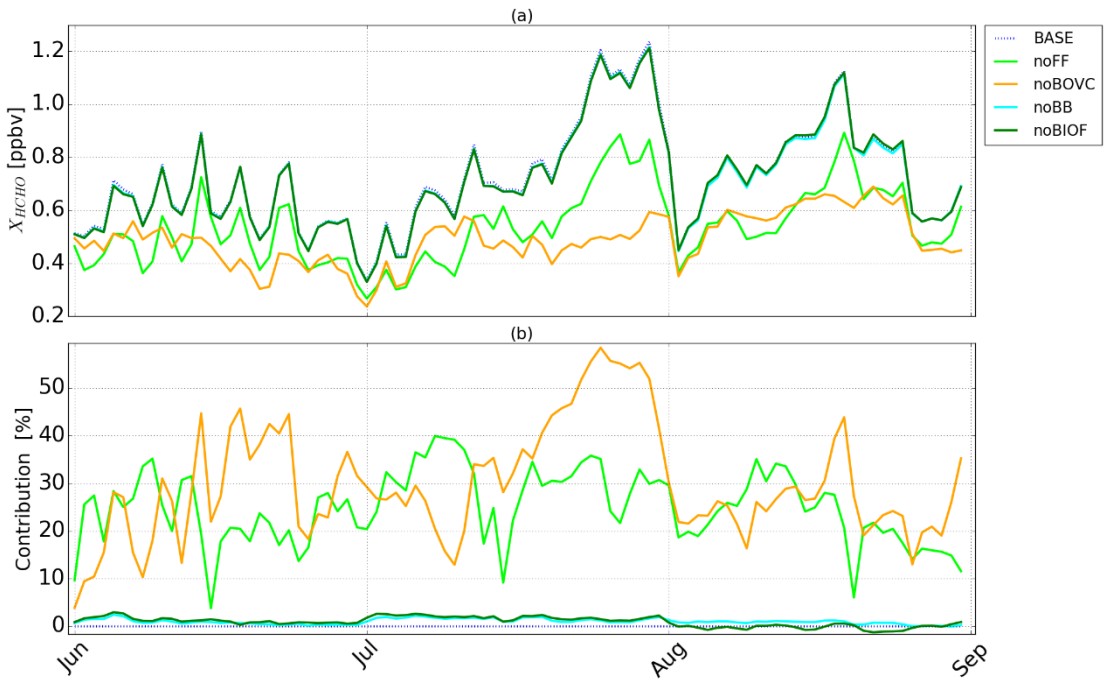

Fig. 7. (a) Daily mean $X_{HCHO}$ (in ppbv) time series averaged over the summertime of 2015 to 2019 above Hefei
simulated by GEOS-Chem, according to the BASE and sensitivity (i.e., noFF, noBVOC, noBB, and noBIOF)
simulations. In the sensitivity simulations, the fossil fuel, biogenic, biomass burning and biofuel emissions of all
atmospheric compounds have been shut off globally, while the CH₄ concentrations are still derived from NOAA
measurements, as for the BASE simulation. (b) Relative contribution of each emission category calculated as the
relative difference between the BASE simulation and the corresponding sensitivity simulation (in %).

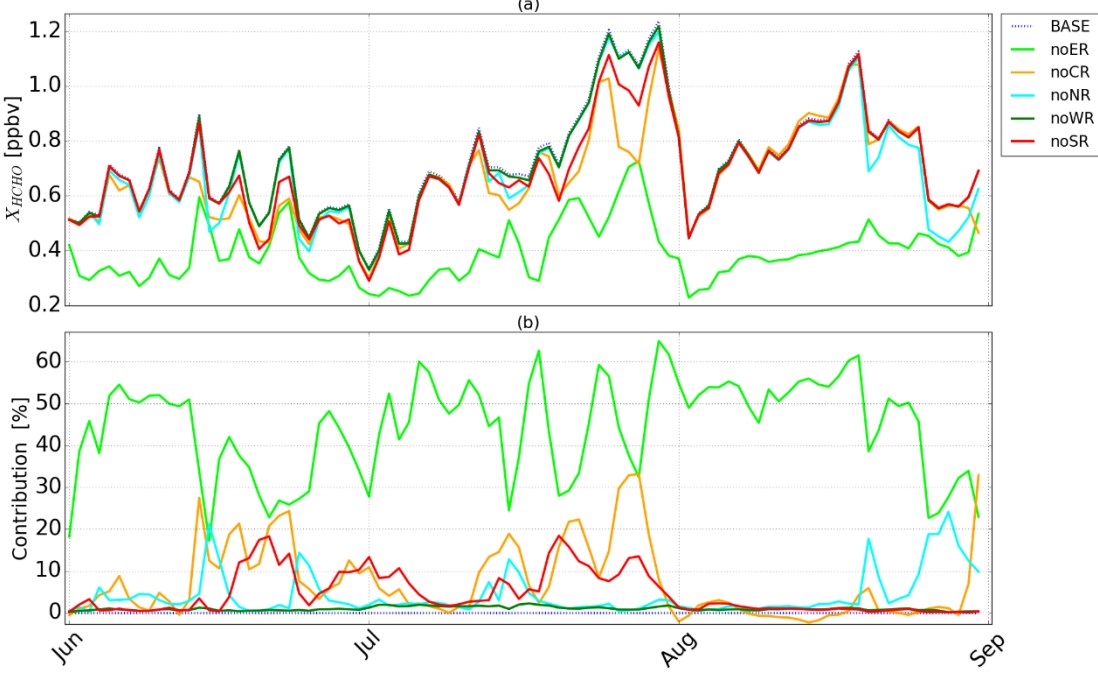

Fig. 8. (a) Daily mean $X_{HCHO}$ (in ppbv) time series averaged over the summertime of 2015 to 2019 above Hefei
simulated by GEOS-Chem, according to the BASE and sensitivity (i.e., noER, noCR, noNR, noWR, and noSR)
simulations. In the sensitivity simulations, the fossil fuel, biogenic, biomass burning and biofuel emissions of all
atmospheric compounds within the studied region have been shut off, while the $CH_4$ concentrations are still derived
from NOAA measurements, as for the BASE simulation. (b) Relative contribution of each geographical region
calculated as the relative difference between the BASE simulation and the corresponding sensitivity simulation
(in %).

# Appendix

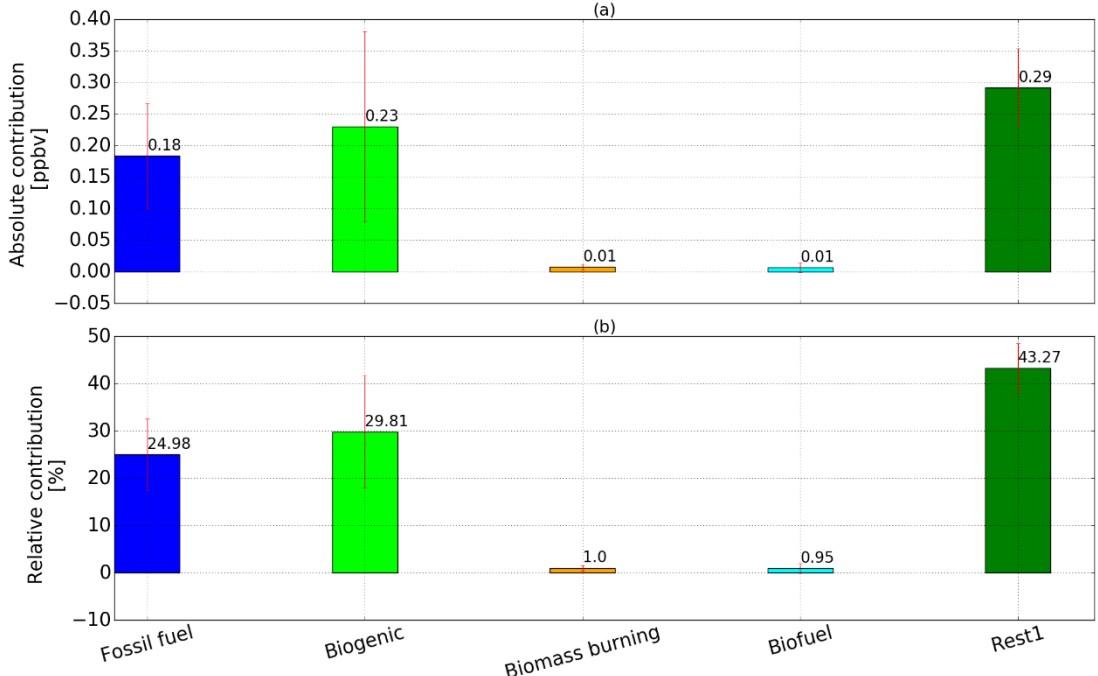

Fig. A1. Absolute (a) and relative (b) contributions of fossil fuel, biogenic, biomass burning, and biofuel emission sources to the observed HCHO summertime enhancements from 2015 to 2019 over Hefei, eastern China. Vertical error bars represent 1-σ standard variation. The remaining contribution (Rest1) was calculated as the difference between the BASE simulation and the sum of all emission category sensitivity simulations.

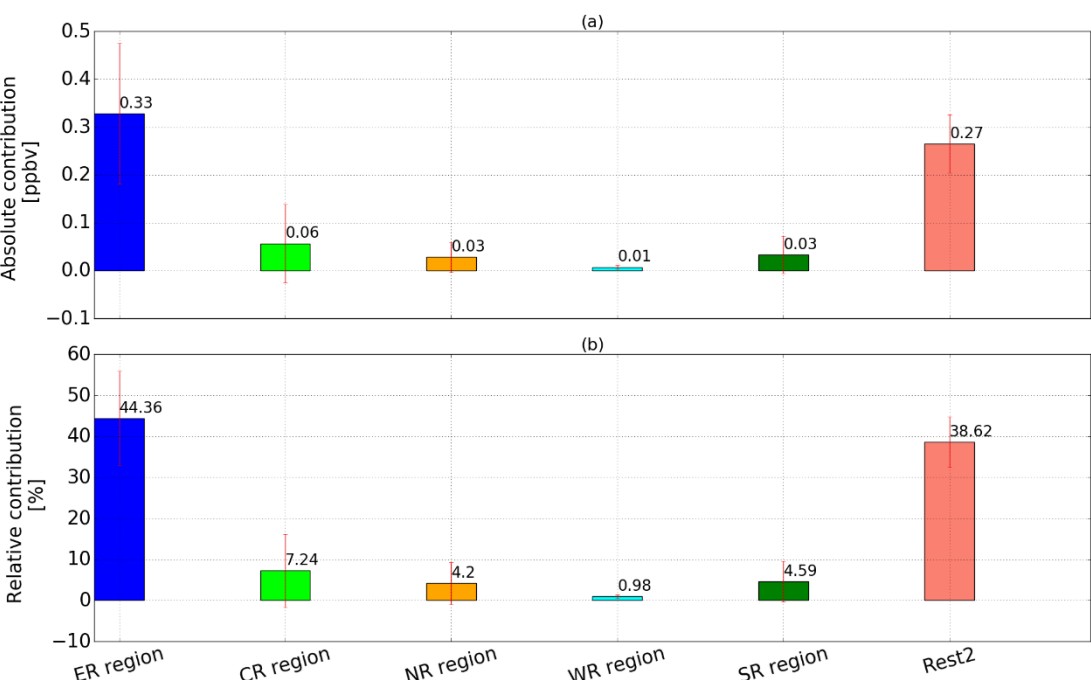

Fig. A2. Absolute (a) and relative (b) contributions of the emission clusters in ER, CR, NR, WR, and SR regions to the observed HCHO summertime enhancements from 2015 to 2019 over Hefei, eastern China. Geographical delimitations for these regions are summarised in Table 3. Vertical error bars represent 1-σ standard variation. The remaining contribution (Rest2) was calculated as the difference between the BASE simulation and the sum of all geographical sensitivity simulations.