# Peer review of "Mapping the drivers of formaldehyde (HCHO) variability from 2015 to 2019 over eastern China: insights from FTIR observation and GEOS-Chem model simulation"

_Atmospheric Chemistry and Physics, 2020_

## Referee Comment (RC1) · Anonymous Referee #2 · 4 Sep 2020

Review of "Mapping the drivers of formaldehyde (HCHO) variability from 2015-2019 over eastern China: insights from FTIR observation and GEOS-Chem model simulation" (acp-2020-544) by Youwen Sun et al.

This study presents and analyses a 5-year time series of HCHO measurements obtained from ground-based FTIR spectra recorded between 2015 and 2019 at Hefei, eastern China. A statistical model is adjusted in order to reproduce the HCHO abundance and variability based on ground level in situ measurements of CO and Ox (O3 + NO2) taken in the vicinity of the FTIR site. CO and Ox are used as tracers for emitted

and photochemical HCHO, respectively, in order to decipher the contribution of direct emissions and oxidation of gas precursors to the HCHO abundance and variability. Estimates of OH radical production from the photolysis of HCHO at the measurement site are also obtained. Finally, GEOS-Chem model simulations are performed to investigate the contribution of different emission categories and geographical regions in China to the HCHO summertime enhancement captured at the measurement site.

This manuscript is well structured and its topic fits the scope of ACP. However, I have several major concerns that prevent me from recommending this work for publication. Those are summarised here below and detailed in my general comments.

- First, this study presents very little novelty compared to the already abundant literature on HCHO. Indeed, just in the past decade there has been a flurry a major advancements as to the observations of atmospheric HCHO, including extensive studies with various instruments on various platforms and modelling studies (sometimes driven by satellite) providing estimates of the HCHO sources worldwide. To my point of view, since the interpretation of the GEOS-Chem results is quite vague, the only novelty would be the new ground-based FTIR observations of HCHO.

- This is the second issue, because the retrievals are poorly characterized and the dataset is not fully exploited. In light of the recent multi-site studies (e.g., Vigouroux et al., 2018), I do not know if presenting an additional FTIR time series, obtained using a retrieval method from another paper, is enough to justify a publication. - The datasets (from both FTIR and GEOS-Chem) are not fully exploited. Indeed, the study lacks overall discussion of the results and presents few perspectives in relation with the literature. Therefore, this gives to the reader a feeling of "unfinished work".

- I have major concerns as to the use of the datasets. In particular, surface and tropospheric VMRs of HCHO are produced from the FTIR retrievals and then used for analysis. However, these retrievals simply do not provide nor contain the necessary information to produce such datasets. As to GEOS-Chem, the description of the model

runs are incomplete and the authors do not discuss some drawbacks, which makes the interpretation of the results difficult and inconclusive.

- Finally, the conclusions and perspectives that are presented in this study for the whole eastern China are actually derived from analysis made at one site only, which tends to really weaken these conclusions. Therefore, I believe that the current manuscript does not befit the standards of ACP and I fear that the work needed to improve it will be too important for a simple revision step.

General comments

- There is little information on the FTIR retrievals of HCHO used in this work and on the characterization of the retrieved product. For instance, a comprehensive error budget is missing (with the systematic and random error terms). Also, considering that the retrieved information for HCHO is quite small (DOFS close to 1), I am wondering to which extent the retrieved profile is affected by the a priori profile in less favourable observational conditions, e.g., around noontime when the probed atmosphere is thinner, or in winter when HCHO is less abundant. For the analyses made with the HCHO time series, it is important to know if the HCHO measurements are biased during certain periods due to a larger influence from the a priori.

- I am very sceptical on the calculation of a mean tropospheric HCHO VMR that has been averaged between the surface and 10 km altitude. It looks like it has been selected arbitrarily. From the DOFS in Fig. 1, we know that the HCHO retrievals provide only one piece of information (DOFS = 1), which is actually the total column. To my understanding, a DOFS of 1 means that basically you have no information at all on how the HCHO VMR profile is distributed vertically and on how it varies. Therefore, this should not allow the extraction of independent information between the surface and an arbitrary selected levels. The only variable that can be used reliably for further data analysis should be the HCHO total column.

- The manuscript lacks sufficient explanation on the time scales and temporal resolution

of the different datasets investigated. For instance, a regression model for source separation is applied to reproduce the observed HCHO based on in situ CO and Ox measurements. However, it is not said if hourly or daily mean in situ measurements were used, and if the model was adjusted to the individual FTIR observations or daily averages. Considering the high intra-day reactivity of HCHO and Ox, any shifts in time (even a few hours) between the datasets might introduce large biases. Again, the manuscript does not say if hourly or daily mean GEOS-Chem outputs are compared to individual or daily mean FTIR measurements, which – by definition – are performed during daytime only.

- Section 2.4: I am wondering to which extent the results of the source separation are influenced by the fact that tropospheric averaged HCHO VMRs are approximated by surface measurements. In situ measurements are significantly affected by local conditions (e.g., vicinity of a major pollution source), whereas tropospheric averaged HCHO VMRs are already more representative of tropospheric chemistry and are also driven by air masses transported from other regions. In the free troposphere, HCHO production from CH4 should play a key role as well, but it can hardly be accounted for by the in situ tracers. There is no discussion nor evaluation of that point.

- Overall, the study reads as an unfinished work because the results are briefly presented and there is not real discussion nor comparison with the literature on observation and modelling of HCHO. It is the case, for example, in Section 3.2, where there is no discussion on what drives the temporal variability of HCHO (e.g., in comparison with other Northern Hemisphere mid-latitude sites), or in Section 3.3, where it is not discussed what could explain the relative contribution of the emitted, photochemical and background HCHO. Similarly, there are many missing information on the GEOS-Chem simulations, e.g.: What are the species whose emissions are shut off? What is considered as being background HCHO? What is contributing to the direct HCHO emissions at Hefei? How about the contribution of CH4, a major and ubiquitous precursor of HCHO?

- Section 3.2: The authors are prompt to conclude that GEOS-Chem can be used with confidence for further analysis, while I find the model evaluation to be quite succinct and inconclusive. For example, the manuscript does not indicate if daily means are used for the comparison with FTIR data or hourly model data collocated in time with the individual FTIR measurements. In p8, lines 9-10, it is stated that the daily and seasonal variability can be reproduced by the model. As to the daily variability, it is impossible to see if in Fig. 3 GEOS-Chem is really able to follow the observed day-to-day variability. As to the seasonality, the model overestimates the HCHO abundance in winter, but it is not discussed explicitly.

- Section 3.2: The diurnal modulation of HCHO has already been investigated several times with remote sensing data, e.g., with spaceborne (De Smedt et al., 2015), ground-based FTIR (Vigouroux et al., 2018; Franco et al., 2016) and MAX-DOAS (Peters et al., 2012) observations. In these studies, the typical diurnal modulation of HCHO at mid-latitudes shows a pronounced peak in the early afternoon, when the photochemistry is enhanced. Moreover, several of these studies showed that global models (including GEOS-Chem) are unable to reproduce the observed modulation. Here, investigating the diurnal variations of HCHO, at a specific site, with a coarse-resolution global model such as GEOS-Chem, has very little meaning. Moreover, GEOS-Chem results are not convincing at all since they do not look to be consistent with the Ox in situ measurements that exhibit a peak in the early afternoon (Fig. 4), and also because the entire diurnal modulation is included in the error bars. Hence, no conclusion can be drawn from this exercise.

- De Smedt et al. (2015) Diurnal, seasonal and long-term variations of global formaldehyde columns inferred from combined OMI and GOME-2 observations, Atmos. Chem. Phys., 15, 12519–12545, doi:10.5194/acp-15-12519-2015.

- Franco et al. (2016) Diurnal cycle and multi-decadal trend of formaldehyde in the remote atmosphere near 46°N, Atmos. Chem. Phys., 16, 4171-4189, doi:10.5194/acp-16-4171-2016.

- Peters et al. (2012) Formaldehyde and nitrogen dioxide over the remote western Pacific Ocean: SCIAMACHY and GOME-2 validation using ship-based MAX-DOAS observations, Atmos. Chem. Phys., 12, 11179–11197, doi:10.5194/acp-12-11179-2012.

- Vigouroux et al. (2018) NDACC harmonized formaldehyde time series from 21 FTIR stations covering a wide range of column abundances, Atmos. Meas. Tech., 11, 5049–5073, doi:10.5194/amt-11-5049-2018.

- Section 3.3: I have to question the reliability of the calculated surface HCHO VMRs. If I understood it well, tropospheric HCHO VMRs derived from FTIR measurements – which were already scaled once to obtain these averaged tropospheric VMRs – are used to produce ground level VMRs of HCHO via a second scaling. First, it is not explained how the scaling was performed. This "double" (or even simple) scaling likely generated large uncertainties on the obtained ground level VMRs. Moreover, decipher-ing surface VMRs of a highly reactive species such as HCHO from ground-based FTIR retrievals with no vertical information (DOFS = 1) and a clear lack of sensitivity in the lowermost layers (Fig. 1) is, to my point of view, not reliable. Hence it casts doubts on the OH production that is deduced from the ground level HCHO VMRs.

- Section 4: Shutting off entire sectors of emissions in global model simulations is rel-atively "dangerous" and has some feedbacks on the modelled species that are difficult to interpret. For instance, turning off the anthropogenic emissions induces significantly lower atmospheric concentrations in NMVOCs, which mainly react with OH. This re-sults in higher concentrations in OH available for the oxidation of other precursors of HCHO, such as CH4, which eventually enhances the HCHO production from the other sources. In case the NO emissions are also suppressed when shutting off an emission inventory (which is the case I think), what is the impact of the missing NO emissions on the overall HCHO burden since NO plays a key role in both HCHO formation (via the degradation of peroxy radicals) and loss (NO contributes to the recycling of OH)? Therefore, I am wondering to which extent the results from the GEOS-Chem simulations can be impacted by such feedbacks and how they can be interpreted.

Specific comments

- Abstract: I think that the abstract should reflect that measurements from only one FTIR site (Hefei) was used for investigating the HCHO variability.

- We can certainly not talk about "trend of HCHO" when it is derived from a 5-year time series only, nor present the current data as representative for real HCHO trends. At best, we can talk about "recent rate of change", with the necessary caveats. Linear regressions adjusted to such a short time series can easily be steered up or down due to exceptionally high or low HCHO levels during a specific season.

- It is repeated several times in the manuscript that the study "should help to improve urban air quality and contributes to the formation of new Chinese clean air policies". It sounds a bit like overselling and a shortcut between scientific works and political decision. I would rather formulate it in a more general way, e.g., "Understanding the sources of VOCs is a necessary step for tackling the problems of poor air quality in eastern China and mitigating the emissions of pollutants."

- p3: I find the first two paragraphs to be quite long while they bring very little useful information to the manuscript, and hence they are not necessary. They can easily be summarised as follows: "The relative contribution of emitted and photochemical sources to atmospheric HCHO has been analysed by using the CO-O3, CO-Ox or CO-CHOCHO tracer pair in various polluted environments (references). In those studies, tropospheric HCHO column measurements were sometimes used as representative of near-surface . . . ."

- I disagree with the authors' statement that "the OH radical production rate from HCHO photolysis estimated in this study provides an evaluation of regional photochemical capacity over eastern China". It gauges only part of the OH production rate since OH in the atmosphere is also produced via many other pathways. That statement must be

tempered.

- p4, lines 19-20: Does this average include the days with no measurements? How many days in a year do the FTIR observational operations represent?

- p4, lines 27-8: What type of model simulation is this? What do you mean by "WACCM special run"? In Vigouroux et al. (2018), I understand it as being a long-term standard model run from 1980 to 2020, providing a kind of climatology for various trace gases.

- p4, line 34: Do you mean that you fit the ILS during the retrieval process?

- p4, lines 36-37: What is the percentage of measurements you excluded this way?

- p6, lines 14-17: I do not see the point here. Did you split your dataset into subsets? Below, in the same paragraph, it is quoted that "all measurements were grouped by months". Please specify if your analysis has been performed on a monthly basis.

- Section 2.5: What are the species whose emissions are typically suppressed when an emission inventory is turned off? Does it affect the NMVOCs only, or the NO emissions as well? How about species such as CH4 and CO? Usually, in global model simulations, the CH4 fields are prescribed, and hence cannot be turned off the same way as for the emission inventories.

- p7, lines 2-3: Could you be more specific as to what is included in "biofuel emissions"? "Fossil fuel emissions" look to include already most of the anthropogenic emissions.

- p7, lines 6-13: Is the delimitation of the geographical regions arbitrary or based on specific criteria? Do such regions present different characteristics in terms of population density, presence of polluting industries, agriculture, surface coverage (forests, deserts. . .)?

- p7, line 14: From what I knew about GEOS-Chem, the tagged simulations did not implement the full chemistry. The main fields were prescribed and originated from previous standard (full chemistry) runs. The tagged simulations included instead a

much reduced chemistry of the tagged tracer (basically, the tracer sinks). Is it the case with your tagged simulations?

- Section 3.1: Are the model profiles smoothed by the averaging kernels of individual observations or by a mean of AvKs calculated from multiple observations? Fig. 3 (top panel) displays model data even when there are no FTIR data available. Where do the AvKs used to smooth such profiles come from?

- p8, lines 15-22: I am not entirely convinced by the explanation. If the dilution of the model information inside a grid box was the only issue, we would basically observe an almost "flat" seasonal cycle. Here, GEOS-Chem performs quite well in the summer enhancements while there is a problem mainly in the winter troughs.

- Section 3.2: The interannual variability in the FTIR time series is mentioned several times, but it is actually not discussed. Looking at Fig. 3, it even seems that there is no interannual variability.

- p10, line 25: Where is located the CRDS analyser? At the site where the Ox in situ measurements are performed?

- Section 4.1: It sounds a bit awkward to state that anthropogenic emissions contribute to summertime HCHO enhancement, since usually the anthropogenic emissions are relatively constant throughout the year. Could you explain how they contribute to an enhancement of HCHO in summer? Isn't it just the photochemistry that is simply enhanced?

- p12, lines 39-41: The manuscript does not present any evidence for this. There should be at least references to the rates of change of CH4 and various NMVOCs that are reported in the literature over the same time period.

Typos/errors

- p1, line 39: I think that "FTIR spectroscopy" is more appropriate than "spectrometry"

- p2, line 6: from 2015 to 2019

- p4, line 32: De Mazière

- p6, line 24: YRD is not defined yet

- p11, line 12: accounted

- p11, line 33: Modelled HCHO was decreased by . . .

- p11, line 37: As a short-lived species

- Please check all your references. For example, Vigouroux et al. (2018) is missing.

---

## Referee Comment (RC2) · Anonymous Referee #1 · 27 Oct 2020

This paper presented a long term observation of formaldehyde from 2015 to 2019 by ground-based high-resolution FTIR at Hefei, eastern China. They used the dataset to assess the performance of the GEOS-Chem model simulation for the specifics of polluted regions over eastern China. The intercomparison of HCHO VMRs between FTIR and GEOS-Chem showed a good agreement; the seasonal and annual trends of HCHO are also well characterized. The tropospheric HCHO VMR from 2015 to 2019 kept increasing. The tracers from ground observation were used to characterize the source of HCHO. Estimates of OH radical production from the photolysis of HCHO

at the measurement site are also obtained. Finally, contributions of various emission sectors and geographical transport to the observed HCHO summertime enhancements are determined using a GEOS-Chem sensitivity study. Overall, this manuscript is well structured, written, and its topic fits the scope of ACP. I recommend this paper for publication subjects to some minor comments.

General comments. 1. Section 2: Characterizations for the retrieval and error analysis should be more detailed. Though ACP mainly focuses on general significance to atmospheric science rather than technical research, a more detailed description for verifying the reliability of the data is needed.

2. The statistics of the GEOS-Chem emission inventories are used to interpreting the HCHO variability. More analysis of data and evidence is needed. There should be at least references to the rates of change of CH4 and various NMVOCs reported in the literature over the same time periods.

Specific comments for correction.

1. Abstract: The abstract should reflect that measurements from only one FTIR site (Hefei) were used for investigating the HCHO variability.

2. Page 3: The first two paragraphs to be quite long while they bring very little useful information to the manuscript, and hence they are not necessary. A concise description is suggested.

3. Page6, line 24: YRD should be defined.

4. Page 10, line 25: Where is the location of the CRDS analyzer? At the site where the Ox in situ measurements is performed?

5. Page 11, line 12: accounted

6. Page11, line 33: Modelled HCHO was decreased by. . .

7. Page11, line 37: As a short-lived species

---

## Author Comment (AC1) · 29 Dec 2020

Response to Referee #1:

Thanks very much for your comments, suggestions and recommendation with respect to improve this paper. The response to all your comments are listed below. There was an extensive discussion among the authors regarding how to revise the content, and this paper is subjected to a major revision for addressing the criticisms by all the referees. Thus, the response is delayed, and we are sorry for this.

Review of "Mapping the drivers of formaldehyde (HCHO) variability from 2015-2019 over eastern China: insights from FTIR observation and GEOS-Chem model simulation" (acp-2020-544) by Youwen Sun et al. This study presents and analyses a 5-year time series of HCHO measurements obtained from ground-based FTIR spectra recorded between 2015 and 2019 at Hefei, eastern China. A statistical model is adjusted in order to reproduce the HCHO abundance and variability based on ground level in situ measurements of CO and Ox ($O_3$ + $NO_2$) taken in the vicinity of the FTIR site. CO and Ox are used as tracers for emitted and photochemical HCHO, respectively, in order to decipher the contribution of direct emissions and oxidation of gas precursors to the HCHO abundance and variability. Estimates of OH radical production from the photolysis of HCHO at the measurement site are also obtained. Finally, GEOS-Chem model simulations are performed to investigate the contribution of different emission categories and geographical regions in China to the HCHO summertime enhancement captured at the measurement site.

This manuscript is well structured and its topic fits the scope of ACP. However, I have several major concerns that prevent me from recommending this work for publication.

**Response:** This paper has been subjected to a major revision based on the comments from two referees. All your comments are appreciated and have been addressed in the revised version. Please see our point by point response as follows.

Main changes/improvements are listed as follows:

**1)** We have included a new section (i.e., section 2.1) to describe the FTIR site and all involved instruments.

**2)** We have included a comparison with the ground level LP-DOAS measurements to ensure that HCHO column measurements at Hefei can be used as representative of near-surface conditions. Please see section 3.1 for details.

**3)** We have included a new section (i.e., section 4.4) to discuss potential factors that drive interannual variability of HCHO over Hefei.

**4)** Furthermore, we have included fully characterizations of the retrievals (section 2.2) and the resulting new dataset is exploited extensively in the revised version. Meanwhile, we include extensive discussion of the results and present sufficient

perspectives in relation with the literature. For instance, we use the statistical results of the emission inventories used in GEOS-Chem sensitivity simulations to explain the observed variability of HCHO (section 2.5). The HCHO observations over Hefei are compared with an unprecedented harmonized HCHO total column dataset from 21 ground-based FTIR stations around the globe provided by Vigouroux et al. (2018) (section 3.2). Source attribution with GEOS-Chem sensitivity simulations are also compared with those of Franco et al. (2016) (section 4).

**5)** The whole paragraph regarding investigation of diurnal modulation of HCHO at Hefei with GEOS-Chem simulations is totally removed.

**6)** Model evaluation has been shifted to the section for source attribution.

Those are summarised here below and detailed in my general comments.

- First, this study presents very little novelty compared to the already abundant literature on HCHO. Indeed, just in the past decade there has been a flurry a major advancements as to the observations of atmospheric HCHO, including extensive studies with various instruments on various platforms and modelling studies (sometimes driven by satellite) providing estimates of the HCHO sources worldwide. To my point of view, since the interpretation of the GEOS-Chem results is quite vague, the only novelty would be the new ground-based FTIR observations of HCHO.

**Response:** In addition to the novelty regarding the new ground-based FTIR observations of HCHO, this study first using FTIR dataset to separate different sources of HCHO and evaluate hydroxyl (OH) radical production rates from HCHO photolysis over China. Meanwhile, in the revised version, contributions of various emission categories and geographical regions in China to the observed HCHO summertime enhancements were analyzed in detail by using a series of GEOS-Chem sensitivity simulations. Furthermore, the revised version presents sufficient perspectives in relation with the literature.

- This is the second issue, because the retrievals are poorly characterized and the dataset is not fully exploited. In light of the recent multi-site studies (e.g., Vigouroux et al., 2018), I do not know if presenting an additional FTIR time series, obtained using a retrieval method from another paper, is enough to justify a publication. - The datasets (from both FTIR and GEOS-Chem) are not fully exploited. Indeed, the study lacks overall discussion of the results and presents few perspectives in relation with the literature. Therefore, this gives to the reader a feeling of "unfinished work".

**Response:** First, we include fully characterizations of the retrievals and the resulting new dataset is exploited extensively in the revised version. In addition, we include extensive discussion of the results and present sufficient perspectives in relation with

the literature. For instance, we use the statistical results of the emission inventories used in GEOS-Chem sensitivity simulations to explain the observed variability of HCHO. The HCHO observations over Hefei are compared with an unprecedented harmonized HCHO total column dataset from 21 ground-based FTIR stations around the globe provided by Vigouroux et al. (2018). Source attribution with GEOS-Chem sensitivity simulations are also compared with those of Franco et al. (2016).

- I have major concerns as to the use of the datasets. In particular, surface and tropospheric VMRs of HCHO are produced from the FTIR retrievals and then used for analysis. However, these retrievals simply do not provide nor contain the necessary information to produce such datasets. As to GEOS-Chem, the description of the model runs are incomplete and the authors do not discuss some drawbacks, which makes the interpretation of the results difficult and inconclusive.

**Response:** The typical DOFS over the total atmosphere obtained at Hefei for HCHO is $1.2 \pm 0.2$ (1σ), meaning that we cannot provide more than one piece of information on the vertical profile. This is the reason that only total columns of HCHO or column-averaged dry air mole fractions of HCHO ($X_{HCHO}$) are discussed in this paper and not vertical profiles. However, HCHO is a tropospheric gas and has a vertical distribution that is heavily weighted toward the lower troposphere. In the revised version, the FTIR $X_{HCHO}$ measurements are compared with the LP-DOAS ground level HCHO measurements. The temporal difference between FTIR and LP-DOAS dataset is within $\pm 5$ minutes. The results show that the HCHO variability observed by FTIR and LP-DOAS are in good agreement with a correlation coefficient ($r^2$) of 0.88. The amplitude of the LP-DOAS ground level measurements is on average 7.89 times that of the FTIR column-averaged measurements. This means HCHO column measurements at Hefei can be used as representative of near-surface conditions. As a result, this study used a constant factor of 7.89 to scale the column-averaged HCHO concentration to ground level HCHO concentration, or vice versa.

As to GEOS-Chem, we include complete descriptions of the GEOS-Chem model runs and discuss some drawbacks of the simulation (e.g., nonlinear effect) in the revised version. We believe the interpretation of the results in the new revision is clear and conclusive.

Please see section 2.5 and 3.1 in the revised version for details.

- Finally, the conclusions and perspectives that are presented in this study for the whole eastern China are actually derived from analysis made at one site only, which tends to really weaken these conclusions. Therefore, I believe that the current manuscript does not befit the standards of ACP and I fear that the work needed to

improve it will be too important for a simple revision step.

**Response:** In the revised version, we have stated that the analysis is based on HCHO measurements at the FTIR site located in Hefei, eastern China. In addition, this paper is subjected to a major revision for addressing the criticisms by all referees. We hope the new version befits the standards of ACP, but if not, we will be very grateful if the referees can sort out what we should do in next step.

General comments

- There is little information on the FTIR retrievals of HCHO used in this work and on the characterization of the retrieved product. For instance, a comprehensive error budget is missing (with the systematic and random error terms). Also, considering that the retrieved information for HCHO is quite small (DOFS close to 1), I am wondering to which extent the retrieved profile is affected by the a priori profile in less favourable observational conditions, e.g., around noontime when the probed atmosphere is thinner, or in winter when HCHO is less abundant. For the analyses made with the HCHO time series, it is important to know if the HCHO measurements are biased during certain periods due to a larger influence from the a priori.

**Response:** In the revised version, we have included detailed description of the FTIR retrievals at Hefei and presented fully characterizations of the retrievals, e.g., a comprehensive error budget is included.

The vertical information contained in the FTIR retrievals can be characterized by the averaging kernel matrix **A** (Rodgers, 2000). The rows of **A** are the so called averaging kernels and they represent the sensitivity of the retrieved profile to the real profile. The area of averaging kernels represents sensitivity of the retrievals to the measurement. It indicates, at each altitude, the fraction of the retrieval at each altitude that comes from the measurement rather than from the *a priori* information (Rodgers, 2000). A value close to zero at a certain altitude indicates that the retrieved profile at that altitude is nearly independent of measurement and is therefore approaching the *a priori* profile. The trace of the averaging kernel matrix **A** is the so called degrees of freedom for signal (DOFS) and it quantifies the number of independent information in the retrieved vertical profile. The ground-based FTIR measurements of HCHO at Hefei have a sensitivity larger than 0.5 from the ground to about 15 km altitude (Fig. 2(a)), indicating that the retrievals are mainly sensitive to the troposphere. This also means that the retrieved profile information above 15 km comes for less than 50% from the measurement, or in other words, that the *a priori* information influences the retrieval by more than 50%. The FTIR measurements taken with a solar intensity variation (SIV) of larger than 10% or retrievals with DOFS of less than 0.7 or root-mean-square (RMS) of fitting residuals of larger than 2% were excluded in this study. This filter criterion excluded the measurements seriously affected by instable weather conditions or by the a priori profile due to low measurement information content in less favourable observational conditions, e.g., around noontime when the

probed atmosphere is thinner, or in winter when HCHO is less abundant. With this criterion, 12.4% of FTIR measurements were excluded in this study. As a result, all measurements used in this study should more come from the measurement rather than the *a priori* information. Please see section 2.2 and the first paragraph of section 3 for details.

- I am very sceptical on the calculation of a mean tropospheric HCHO VMR that has been averaged between the surface and 10 km altitude. It looks like it has been selected arbitrarily. From the DOFS in Fig. 1, we know that the HCHO retrievals provide only one piece of information (DOFS = 1), which is actually the total column. To my understanding, a DOFS of 1 means that basically you have no information at all on how the HCHO VMR profile is distributed vertically and on how it varies. Therefore, this should not allow the extraction of independent information between the surface and an arbitrary selected levels. The only variable that can be used reliably for further data analysis should be the HCHO total column.

**Response:** The typical DOFS over the total atmosphere obtained at Hefei for HCHO is $1.2 \pm 0.2$ ($1\sigma$), meaning that we cannot provide more than one piece of information on the vertical profile. However, HCHO is a tropospheric gas and has a vertical distribution that is heavily weighted toward the lower troposphere. The HCHO concentration decreased by 72.7% with an increase in the height from surface to 3 km and continued to decrease slowly in the troposphere above 3 km. The HCHO partial column below 3 km accounted for 67.1% of HCHO total column. This percentage is expected to show less seasonal variation since the shape of the retrieved profile is very similar to the shape of the a priori profile due to the low DOFS. For above reason, in the ACPD version, we taken the partial layer between the surface and 10 km altitude as a whole and used it for all analyses. This selected partial layer basically holds all of the total DOFS and thus can be used reliably. However, in the revised version, we used only total columns of HCHO or column-averaged dry air mole fractions of HCHO ($X_{HCHO}$) to avoid misleading, though using $X_{HCHO}$ and tropospheric-averaged dry air mole fractions of HCHO (between the surface and 10 km) basically result in the same conclusion.

- The manuscript lacks sufficient explanation on the time scales and temporal resolution of the different datasets investigated. For instance, a regression model for source separation is applied to reproduce the observed HCHO based on in situ CO and Ox measurements. However, it is not said if hourly or daily mean in situ measurements were used, and if the model was adjusted to the individual FTIR observations or daily averages. Considering the high intra-day reactivity of HCHO and Ox, any shifts in time (even a few hours) between the datasets might introduce

large biases. Again, the manuscript does not say if hourly or daily mean GEOS-Chem outputs are compared to individual or daily mean FTIR measurements, which − by definition – are performed during daytime only.

**Response:** We have included sufficient explanation on the time scales and temporal resolution of the different datasets in the revised version. The LP-DOAS ground level HCHO measurements nearest to each individual FTIR $X_{HCHO}$ measurement were included for comparison. The temporal difference between FTIR and LP-DOAS dataset is within ± 5 minutes. The seasonality and interannual variability of HCHO from 2015−2019 are determined by using the bootstrap resampling method of Gardiner et al. (2008) with a 3$^{rd}$ Fourier series plus a linear function to fit FTIR daily mean time series of $X_{HCHO}$. The CNEMC ground level CO and O$_x$ measurements nearest to each individual FTIR $X_{HCHO}$ measurement were included for source separation. The temporal difference between FTIR and CNEMC dataset is within ± 30 minutes. For the ground level H$_2$O and O$_3$ datasets used in estimation of the OH production rates, only measurements nearest to each individual FTIR measurement were considered. The temporal difference between FTIR and CNEMC (CRDS) is within ± 30 minute (± 30 second). Both daily and monthly means of GEOS-Chem outputs are compared to the concurrent FTIR measurements, which are performed for the days with available FTIR observations only.

- Section 2.4: I am wondering to which extent the results of the source separation are influenced by the fact that tropospheric averaged HCHO VMRs are approximated by surface measurements. In situ measurements are significantly affected by local conditions (e.g., vicinity of a major pollution source), whereas tropospheric averaged HCHO VMRs are already more representative of tropospheric chemistry and are also driven by air masses transported from other regions. In the free troposphere, HCHO production from CH$_4$ should play a key role as well, but it can hardly be accounted for by the in situ tracers. There is no discussion nor evaluation of that point.

**Response:** In the revised version, the FTIR $X_{HCHO}$ measurements are compared with the LP-DOAS ground level HCHO measurements. The temporal difference between FTIR and LP-DOAS dataset is within ± 5 minutes. The results show that the HCHO variability observed by FTIR and LP-DOAS are in good agreement with a correlation coefficient ($r^2$) of 0.88. The amplitude of the LP-DOAS ground level measurements is on average 7.89 times that of the FTIR column-averaged measurements. This means HCHO column measurements at Hefei can be used as representative of near-surface conditions. As a result, this study used a constant factor of 7.89 to scale the column-averaged HCHO concentration to ground level HCHO concentration, or vice versa.

Over polluted atmosphere, the HCHO column measurements can be used as representative of near-surface conditions because HCHO is a tropospheric gas and has a vertical distribution that is heavily weighted toward the lower troposphere (Martin et al., 2004). As shown in Fig.2(c), the HCHO concentration decreased by 72.7% with an increase in the height from surface to 3 km and continued to decrease slowly in the troposphere above 3 km. The HCHO partial column below 3 km accounted for 67.1% of HCHO total column. This percentage is expected to show less seasonal variation since the shape of the retrieved profile is very similar to the shape of the *a priori* profile due to the low DOFS (Fig. 2 (c)). Many studies have taken advantage of this favorable vertical distribution of HCHO to derive surface emissions of VOCs from space (e.g. Palmer et al., 2003; Millet et al., 2008; Boersma et al., 2009; Stavrakou et al., 2009; Fortems-Cheiney et al., 2012; Barkley et al., 2013; Marais et al., 2014; Streets et al., 2013; Gao et al., 2018). Meanwhile, the use of HCHO column measurements to explore tropospheric $O_3$ sensitivities has been the subject of several past studies, which disclosed that this diagnosis of $O_3$ production rate ($PO_3$) is consistent with the findings of surface photochemistry (eg., Martin et al., 2004; Duncan et al., 2010; Choi et al., 2012; Witte et al., 2011; Jin and Holloway, 2015; Mahajan et al., 2015; Jin et al., 2017; Schroeder et al. 2017). Source separation of atmospheric HCHO in Hong et al. (2018) and Su et al. (2019) also taken the advantage that column measurements of HCHO are fairly representative of near-surface conditions.

It is worth noting that imperfections in source separation with this regression model are likely to become significant in certain cases. In this study, photochemical HCHO production from $CH_4$ oxidation in the free troposphere which can hardly be accounted for by the in situ tracers is in fact erroneously (or at least partly) interpreted background HCHO. In addition, the measurements with large temporal variations of $HCHO/CO$ or $HCHO/O_x$ ratios generally can't be reproduced by this regression model. A more sophisticated multi-regression model might be able to reduce the uncertainties, but this is beyond the scope of present work.

The correlation coefficient value ($r^2$) from the regression analysis indicates the proportion of HCHO measurements that can be reproduced by the regression model (Green, 1998). The results indicate that this proportion is for all subsets of data well above 80%, and up to 92%, reflecting that the $CO-O_x$ tracer pair – while not perfect – generally replicates well the observations.

We have included all above discussion in the revised version.

- Overall, the study reads as an unfinished work because the results are briefly

presented and there is not real discussion nor comparison with the literature on observation and modelling of HCHO. It is the case, for example, in Section 3.2, where there is no discussion on what drives the temporal variability of HCHO (e.g., in comparison with other Northern Hemisphere mid-latitude sites), or in Section 3.3, where it is not discussed what could explain the relative contribution of the emitted, photochemical and background HCHO. Similarly, there are many missing information on the GEOSChem simulations, e.g.: What are the species whose emissions are shut off? What is considered as being background HCHO? What is contributing to the direct HCHO emissions at Hefei? How about the contribution of $CH_4$, a major and ubiquitous precursor of HCHO?

**Response:** In the revised version, we have included extensive discussion of the results and comparisons with the literature on observation and modelling of HCHO. For instance, we use the statistical results of the emission inventories used in GEOS-Chem sensitivity simulations to explain the observed variability of HCHO. The HCHO observations over Hefei are compared with an unprecedented harmonized HCHO total column dataset from 21 ground-based FTIR stations around the globe provided by Vigouroux et al. (2018). Source attribution with GEOS-Chem sensitivity simulations are also compared with those of Franco et al. (2016). As evidenced in Table 2, the emitted HCHO are mainly from fossil fuel and biomass burning emissions. In addition to oxidation of $CH_4$, oxidations of both fossil fuel and biogenic NMVOCs could have large contributions to photochemical HCHO, which are discussed in detail in section 4.2. Background represents the regional HCHO condition in the background atmosphere. For the polluted atmosphere over Hefei, it is impossible to directly measure the background HCHO concentration and thus an empirical value derived previous studies in the YRD region was used. According to the ground level measurements of HCHO at a rural site in the Yangtze River Delta (YRD)  region by Ma et al. (2016) and Wang et al. (2015), the background level of HCHO near the surface was approximately 1.0 ppbv in springtime. When an emission inventory was shut off, global emissions of all atmospheric compounds in this inventory were overwritten with zero. We followed the method of Franco et al. (2016) and did not shut off the $CH_4$ inventory in all sensitivity simulations, i.e., $CH_4$ concentrations were still derived from the NOAA measurements as for the BASE simulation. Please see section 2.5 for GEOS-Chem model configurations in detail.

- Section 3.2: The authors are prompt to conclude that GEOS-Chem can be used with confidence for further analysis, while I find the model evaluation to be quite succinct and inconclusive. For example, the manuscript does not indicate if daily means are used for the comparison with FTIR data or hourly model data collocated in time with

the individual FTIR measurements. In p8, lines 9-10, it is stated that the daily and seasonal variability can be reproduced by the model. As to the daily variability, it is impossible to see if in Fig. 3 GEOS-Chem is really able to follow the observed day-to-day variability. As to the seasonality, the model overestimates the HCHO abundance in winter, but it is not discussed explicitly.

**Response:** We have addressed above criticisms in the revised version. Both daily and monthly means of GEOS-Chem outputs are compared to the concurrent FTIR measurements, which are performed for the days with available FTIR observations only. Fig. 4 (a) shows daily mean time series of $X_{HCHO}$ comparison between the FTIR observation and the smoothed GEOS-Chem model simulation from 2015−2019. Fig. 4 (b) is the comparison in term of seasonal cycle derived from Fig. 4 (a) for the days with available FTIR observations only. The observed day-to-day variability cannot be always reproduced by the GEOS-Chem simulation, especially in the trough and peak of the measurements (Fig. 4(a)). This can be partially explained by the fact that different oxidation pathways of VOCs precursors leading the HCHO production, which are numerous, might not be optimally implemented (especially very short-lived VOCs) or merely not considered in the model (Franco et al., 2016). In addition, large uncertainties remain concerning the various sources of precursor emissions, their geographical distribution and how this latter one can influence the air masses over polluted sites such as Hefei. Finally, GEOS-Chem homogenises HCHO concentration over a large coverage area due to its relatively coarse spatial resolution (here 2°× 2.5°). The Hefei site located in a densely populated and industrialised area in eastern China. The regional difference in HCHO concentration could aggravate the inhomogeneity within the selected GEOS-Chem coverage area, which also affects the comparison with observations. However, the measured feature in term of seasonal cycle of HCHO loadings over Hefei can be reproduced by GEOS-Chem simulations with a correlation coefficient ($r^2$) of 0.78 (Fig. 4(b)).

- Section 3.2: The diurnal modulation of HCHO has already been investigated several times with remote sensing data, e.g., with spaceborne (De Smedt et al., 2015), groundbased FTIR (Vigouroux et al., 2018; Franco et al., 2016) and MAX-DOAS (Peters et al., 2012) observations. In these studies, the typical diurnal modulation of HCHO at midlatitudes shows a pronounced peak in the early afternoon, when the photochemistry is enhanced. Moreover, several of these studies showed that global models (including GEOS-Chem) are unable to reproduce the observed modulation. Here, investigating the diurnal variations of HCHO, at a specific site, with a coarse-resolution global model such as GEOS-Chem, has very little meaning. Moreover, GEOS-Chem results are not convincing at all since they do not look to be

consistent with the Ox in situ measurements that exhibit a peak in the early afternoon (Fig. 4), and also because the entire diurnal modulation is included in the error bars. Hence, no conclusion can be drawn from this exercise.

- De Smedt et al. (2015) Diurnal, seasonal and long-term variations of global formaldehyde columns inferred from combined OMI and GOME-2 observations, Atmos. Chem. Phys., 15, 12519–12545, doi:10.5194/acp-15-12519-2015.

- Franco et al. (2016) Diurnal cycle and multi-decadal trend of formaldehyde in the remote atmosphere near 46°N, Atmos. Chem. Phys., 16, 4171-4189, doi:10.5194/acp-16-4171-2016.

- Peters et al. (2012) Formaldehyde and nitrogen dioxide over the remote western Pacific Ocean: SCIAMACHY and GOME-2 validation using ship-based MAX-DOAS observations, Atmos. Chem. Phys., 12, 11179–11197, doi:10.5194/acp-12-11179-2012.

- Vigouroux et al. (2018) NDACC harmonized formaldehyde time series from 21 FTIR stations covering a wide range of column abundances, Atmos. Meas. Tech., 11, 5049–5073, doi:10.5194/amt-11-5049-2018.

**Response:** In the revised version, the whole paragraph regarding investigation of diurnal modulation of HCHO at Hefei with GEOS-Chem simulations is totally removed. We follow the referee's comments that this exercise contributed trivial but easily mislead reader's current knowledge.

- Section 3.3: I have to question the reliability of the calculated surface HCHO VMRs. If I understood it well, tropospheric HCHO VMRs derived from FTIR measurements –which were already scaled once to obtain these averaged tropospheric VMRs – are used to produce ground level VMRs of HCHO via a second scaling. First, it is not explained how the scaling was performed. This "double" (or even simple) scaling likely generated large uncertainties on the obtained ground level VMRs. Moreover, deciphering surface VMRs of a highly reactive species such as HCHO from ground-based FTIR retrievals with no vertical information (DOFS = 1) and a clear lack of sensitivity in the lowermost layers (Fig. 1) is, to my point of view, not reliable. Hence it casts doubts on the OH production that is deduced from the ground level HCHO VMRs.

**Response:** In the revised version, the FTIR $X_{HCHO}$ measurements are compared with the LP-DOAS ground level HCHO measurements. The temporal difference between FTIR and LP-DOAS dataset is within ± 5 minutes. The results show that the HCHO variability observed by FTIR and LP-DOAS are in good agreement with a correlation coefficient ($r^2$) of 0.88. The amplitude of the LP-DOAS ground level measurements is on average 7.89 times that of the FTIR column-averaged measurements. This means

HCHO column measurements at Hefei can be used as representative of near-surface conditions. As a result, this study used a constant factor of 7.89 to scale the column-averaged HCHO concentration to ground level HCHO concentration, or vice versa. **For each case, we only scale the results once time**.

Over polluted atmosphere, the HCHO column measurements can be used as representative of near-surface conditions because HCHO is a tropospheric gas and has a vertical distribution that is heavily weighted toward the lower troposphere (Martin et al., 2004). As shown in Fig.2(c), the HCHO concentration decreased by 72.7% with an increase in the height from surface to 3 km and continued to decrease slowly in the troposphere above 3 km. The HCHO partial column below 3 km accounted for 67.1% of HCHO total column. This percentage is expected to show less seasonal variation since the shape of the retrieved profile is very similar to the shape of the *a priori* profile due to the low DOFS (Fig. 2 (c)). Many studies have taken advantage of this favorable vertical distribution of HCHO to derive surface emissions of VOCs from space (e.g. Palmer et al., 2003; Millet et al., 2008; Boersma et al., 2009; Stavrakou et al., 2009; Fortems-Cheiney et al., 2012; Barkley et al., 2013; Marais et al., 2014; Streets et al., 2013; Gao et al., 2018). Meanwhile, the use of HCHO column measurements to explore tropospheric $O_3$ sensitivities has been the subject of several past studies, which disclosed that this diagnosis of $O_3$ production rate ($PO_3$) is consistent with the findings of surface photochemistry (eg., Martin et al., 2004; Duncan et al., 2010; Choi et al., 2012; Witte et al., 2011; Jin and Holloway, 2015; Mahajan et al., 2015; Jin et al., 2017; Schroeder et al. 2017). Source separation of atmospheric HCHO in Hong et al. (2018) and Su et al. (2019) also taken the advantage that column measurements of HCHO are fairly representative of near-surface conditions.

- Section 4: Shutting off entire sectors of emissions in global model simulations is relatively "dangerous" and has some feedbacks on the modelled species that are difficult to interpret. For instance, turning off the anthropogenic emissions induces significantly lower atmospheric concentrations in NMVOCs, which mainly react with OH. This results in higher concentrations in OH available for the oxidation of other precursors of HCHO, such as $CH_4$, which eventually enhances the HCHO production from the other sources. In case the NO emissions are also suppressed when shutting off an emission inventory (which is the case I think), what is the impact of the missing NO emissions on the overall HCHO burden since NO plays a key role in both HCHO formation (via the degradation of peroxy radicals) and loss (NO contributes to the recycling of OH)? Therefore, I am wondering to which extent the results from the GEOS-Chem simulations can be impacted by such feedbacks and how they can be

interpreted.

**Response:** In the revised version, our interpretation is as follows. "Indeed, shutting off some emission sources in the GEOS-Chem sensitivity simulations eventually resulted in slightly enhanced HCHO amounts (by 1−1.5 %) compared to the BASE simulation, as shown in Fig. 7(b) for the noBIOF simulation and, to a lesser extent, for the noBB simulation during later summer. In these particular cases, shutting off an emission inventory may induce significantly lower concentrations in many atmospheric compounds globally, some of which mainly react with OH. This would lead to higher OH concentrations available for the oxidation of HCHO precursors, and eventually enhances the HCHO production from other emission categories (Franco et al., 2016). However, it is difficult to quantify the nonlinear impact of each individual emission category, since the types of atmospheric compounds and their concentrations in each emission category are different. Especially when the emissions of NO are suppressed, the impacts become hard to assess, since this compound plays a key role in both HCHO formation (through the degradation of peroxy radicals) and destruction (by contributing to the regeneration of OH) (Franco et al., 2016). Investigating the nonlinear impact of each individual emission category would require additional work that is beyond the scope of the present work."   Please see section 4.2 for details.

Specific comments

- Abstract: I think that the abstract should reflect that measurements from only one FTIR site (Hefei) was used for investigating the HCHO variability.

**Response:** In the revised version, we have stated that the analysis is based on HCHO measurements at the FTIR site located in Hefei, eastern China. Please see abstract in the revised version for detail.

- We can certainly not talk about "trend of HCHO" when it is derived from a 5-year time series only, nor present the current data as representative for real HCHO trends. At best, we can talk about "recent rate of change", with the necessary caveats. Linear regressions adjusted to such a short time series can easily be steered up or down due to exceptionally high or low HCHO levels during a specific season.

**Response:** In the revised version,"trend of HCHO" has been replaced by "change rate of HCHO" throughout the paper.

- It is repeated several times in the manuscript that the study "should help to improve urban air quality and contributes to the formation of new Chinese clean air policies". It sounds a bit like overselling and a shortcut between scientific works and political decision. I would rather formulate it in a more general way, e.g., "Understanding the

sources of VOCs is a necessary step for tackling the problems of poor air quality in eastern China and mitigating the emissions of pollutants."

**Response:** In the revised version, we have modified this description as your suggestion. Please see abstract, section 1, and conclusion in the revised version for detail.

- p3: I find the first two paragraphs to be quite long while they bring very little useful information to the manuscript, and hence they are not necessary. They can easily be summarised as follows: "The relative contribution of emitted and photochemical sources to atmospheric HCHO has been analysed by using the CO-O$_3$, CO-O$_x$ or COCHOCHO tracer pair in various polluted environments (references). In those studies, tropospheric HCHO column measurements were sometimes used as representative of near-surface ……"

**Response:** In the revised version, we have modified these two paragraphs as your suggestion. Please see section 1 in the revised version for detail.

- I disagree with the authors' statement that "the OH radical production rate from HCHO photolysis estimated in this study provides an evaluation of regional photochemical capacity over eastern China". It gauges only part of the OH production rate since OH in the atmosphere is also produced via many other pathways. That statement must be tempered.

**Response:** In the revised version, we have tempered this sentence as your suggestion, i.e., the OH radical production rates from HCHO photolysis estimated in this study provide an evaluation of regional photochemical capacity related to the degradation of HCHO over eastern China. Please see section 1 in the revised version for detail.

- p4, lines 19-20: Does this average include the days with no measurements? How many days in a year do the FTIR observational operations represent?

**Response:** This average did not include the days with no measurements. The near infrared (NIR) and middle infrared (MIR) solar spectra are alternately recorded in routine observations (Wang et al., 2017). To balance the TCCON and NDACC measurements, the number of HCHO measurements on each measurement day varied from 1−17 with an average of 6. In total, there were 523 days of qualified measurements between 2015 and 2019. Please see section 2.1 in the revised version for detail.

- p4, lines 27-8: What type of model simulation is this? What do you mean by

"WACCM special run"? In Vigouroux et al. (2018), I understand it as being a long-term standard model run from 1980 to 2020, providing a kind of climatology for various trace gases.

**Response:** We use it the same as that in Vigouroux et al. (2018), i.e., the *a priori* profiles of other gases were from the averages of the Whole-Atmosphere Community Climate Model version 6 (WACCM) simulations from 1980 to 2020. Please see section 2.2.1 in the revised version for detail.

- p4, line 34: Do you mean that you fit the ILS during the retrieval process?

**Response:** What we mean is " We regularly used a low-pressure HBr cell to diagnose the instrument line shape (ILS) of the high resolution FTIR spectrometer at Hefei and included the measured ILS in the retrieval (Hase et al., 2012; Sun et al., 2018).". Please see section 2.2.1 in the revised version for detail.

- p4, lines 36-37: What is the percentage of measurements you excluded this way?

**Response:** With this criterion, 12.4% of FTIR measurements were excluded in subsequent study. Please see the first paragraph of section 3 for details.

- p6, lines 14-17: I do not see the point here. Did you split your dataset into subsets?

**Response:** Yes, we grouped all measurements by month and performed the regression analysis for source separation on a monthly basis. Please see section 2.3 for details.

Below, in the same paragraph, it is quoted that "all measurements were grouped by months". Please specify if your analysis has been performed on a monthly basis.

**Response:** We have included this statement in the revised version. Please see section 2.3 for details.

- Section 2.5: What are the species whose emissions are typically suppressed when an emission inventory is turned off? Does it affect the NMVOCs only, or the NO emissions as well? How about species such as $CH_4$ and CO? Usually, in global model simulations, the $CH_4$ fields are prescribed, and hence cannot be turned off the same way as for the emission inventories.

**Response:** When an emission inventory was shut off, global emissions of all atmospheric compounds in this inventory were overwritten with zero. We followed the method of Franco et al. (2016) and did not shut off the $CH_4$ inventory in all sensitivity simulations, i.e., $CH_4$ concentrations were still derived from the NOAA measurements as for the BASE simulation. Except $CH_4$, the NMVOCs, NO and CO are all suppressed when shutting off an emission inventory. Please see section 2.5 for

GEOS-Chem model configurations in detail.

- p7, lines 2-3: Could you be more specific as to what is included in "biofuel emissions"? "Fossil fuel emissions" look to include already most of the anthropogenic emissions.

**Response:** In this study, we separated the anthropogenic emission into fossil fuel and biofuel emissions. The global biofuel inventory is only available for the year 2015. The number of atmospheric compounds and the emission amounts in biofuel inventory are much smaller than those in fossil fuel inventory. Indeed, fossil fuel emissions include most of the anthropogenic emissions. Please see section 2.5 for GEOS-Chem model configurations in detail.

- p7, lines 6-13: Is the delimitation of the geographical regions arbitrary or based on specific criteria? Do such regions present different characteristics in terms of population density, presence of polluting industries, agriculture, surface coverage (forests, deserts…..)?

**Response:** The delimitation of these geographical regions is based on the level of urbanization and industrialization in China. Region ① in Fig. 1(a) only covers few sparsely city clusters representing the region with least population and industrialization in China (Lu et al. 2019). Regions ②, ④, and ⑤ cover the North China Plain (NCP), YRD, and Pearl River Delta (PRD) city clusters, respectively, which are the three most developed city clusters with severe air pollution in China. Region ③ covers the Sichuan Basin (SCB) and central Yangtze River (CYR) city clusters with newly emerging severe air pollution in China. Please see section 2.5 for GEOS-Chem model configurations in detail.

- p7, line 14: From what I knew about GEOS-Chem, the tagged simulations did not implement the full chemistry. The main fields were prescribed and originated from previous standard (full chemistry) runs. The tagged simulations included instead a much reduced chemistry of the tagged tracer (basically, the tracer sinks). Is it the case with your tagged simulations?

**Response:** All sensitivity simulations are the same as the standard full chemistry simulation except that certain emissions or emissions within certain specific regions are shut off. As a result, it was wrong to call this kind of simulation as tagged simulation in previous version. However, in the revised version, we call it as sensitivity simulation. Thus, this misleading situation should be avoided. Please see section 2.5 for GEOS-Chem model configurations in detail.

- Section 3.1: Are the model profiles smoothed by the averaging kernels of individual observations or by a mean of AvKs calculated from multiple observations? Fig. 3 (top panel) displays model data even when there are no FTIR data available. Where do the AvKs used to smooth such profiles come from?

**Response:** The interpolated profiles were smoothed by the seasonal mean FTIR averaging kernels and a priori profiles rather than individual ones. As a result, model can display data even when there are no FTIR data available. However, the comparison in term of seasonal cycle only performed for the days with available FTIR observations only. Please see section 4.1 for details.

- p8, lines 15-22: I am not entirely convinced by the explanation. If the dilution of the model information inside a grid box was the only issue, we would basically observe an almost "flat" seasonal cycle. Here, GEOS-Chem performs quite well in the summer enhancements while there is a problem mainly in the winter troughs.

**Response:** In the revised version, the explanation becomes " The observed day-to-day variability cannot be always reproduced by the GEOS-Chem simulation, especially in the trough and peak of the measurements (Fig. 4(a)). This can be partially explained by the fact that many oxidation pathways of VOCs precursors leading the HCHO production, which are numerous, might not be optimally implemented (especially very short-lived VOCs) or merely not considered in the model (Franco et al., 2016). In addition, large uncertainties remain concerning the various sources of precursor emissions, their geographical distributions and how these sources can influence the air masses over polluted sites such as Hefei. Finally, GEOS-Chem averages HCHO concentration over a large coverage area due to its relatively coarse spatial resolution (here $2° \times 2.5°$). The Hefei site is located in a densely populated and industrialised area in eastern China. The regional differences in HCHO concentration could aggravate the inhomogeneity within the selected GEOS-Chem coverage grid cell, which also affects the comparison with observations. Nevertheless, the measured feature in term of seasonal cycle of HCHO loadings over Hefei can be reproduced by GEOS-Chem simulations with a correlation coefficient (r2) of 0.78 (Fig. 4(b))."

- Section 3.2: The interannual variability in the FTIR time series is mentioned several times, but it is actually not discussed. Looking at Fig. 3, it even seems that there is no interannual variability.

**Response:** In the revised version, section 4.4 presents potential factors that drive interannual variability of HCHO over Hefei in detail.

- p10, line 25: Where is located the CRDS analyser? At the site where the $O_x$ in situ

measurements are performed?

**Response:** In the revised version, we have stated that the CRDS analyser located side by side with the FTIR instrument. Please see section 2.1 and Fig. 1(b) for details.

- Section 4.1: It sounds a bit awkward to state that anthropogenic emissions contribute to summertime HCHO enhancement, since usually the anthropogenic emissions are relatively constant throughout the year. Could you explain how they contribute to an enhancement of HCHO in summer? Isn't it just the photochemistry that is simply enhanced?

**Response:** When an emission inventory was shut off, global emissions of all atmospheric compounds in this inventory were overwritten with zero. We followed the method of Franco et al. (2016) and did not shut off the CH$_4$ inventory in all sensitivity simulations, i.e., CH$_4$ concentrations were still derived from the NOAA measurements as for the BASE simulation. As a result, the NMVOCs, NO and CO (except CH$_4$) are all suppressed when shutting off the anthropogenic emission inventories (fossil fuel + biofuel). So anthropogenic contribution here is mainly due to the anthropogenic emitted HCHO and photochemical HCHO from oxidation of anthropogenic NMVOCs. Please see section 4.2 for detailed explanation.

- p12, lines 39-41: The manuscript does not present any evidence for this. There should be at least references to the rates of change of CH$_4$ and various NMVOCs that are reported in the literature over the same time period.

**Response:** In the revised version, we have used the statistical results of the emission inventories used in GEOS-Chem sensitivity simulations to explain the observed variability of HCHO. Section 4.4 presents potential factors that drive interannual variability of HCHO over Hefei in detail.

Typos/errors
- p1, line 39: I think that "FTIR spectroscopy" is more appropriate than "spectrometry"
- p2, line 6: from 2015 to 2019
- p4, line 32: De Mazière
- p6, line 24: YRD is not defined yet
- p11, line 12: accounted
- p11, line 33: Modelled HCHO was decreased by ……
- p11, line 37: As a short-lived species
- Please check all your references. For example, Vigouroux et al. (2018) is missing.

**Response:** All above typos/errors have been corrected in the revised version.

---

## Author Comment (AC2) · 29 Dec 2020

Response to Referee #2:

Thanks very much for your comments, suggestions and recommendation with respect to improve this paper. The response to all your comments are listed below. There was an extensive discussion among the authors regarding how to revise the content, and this paper is subjected to a major revision for addressing the criticisms by all the referees. Thus, the response is delayed, and we are sorry for this.

This paper presented a long term observation of formaldehyde from 2015 to 2019 by ground-based high-resolution FTIR at Hefei, eastern China. They used the dataset to assess the performance of the GEOS-Chem model simulation for the specifics of polluted regions over eastern China. The intercomparison of HCHO VMRs between FTIR and GEOS-Chem showed a good agreement; the seasonal and annual trends of HCHO are also well characterized. The tropospheric HCHO VMR from 2015 to 2019 kept increasing. The tracers from ground observation were used to characterize the source of HCHO. Estimates of OH radical production from the photolysis of HCHO at the measurement site are also obtained. Finally, contributions of various emission sectors and geographical transport to the observed HCHO summertime enhancements are determined using a GEOS-Chem sensitivity study. Overall, this manuscript is well structured, written, and its topic fits the scope of ACP. I recommend this paper for publication subjects to some minor comments.

General comments.

1. Section 2: Characterizations for the retrieval and error analysis should be more detailed. Though ACP mainly focuses on general significance to atmospheric science rather than technical research, a more detailed description for verifying the reliability of the data is needed.

**Response:** We have presented detailed characterizations for the FTIR retrieval and error analysis in the revised version.

2. The statistics of the GEOS-Chem emission inventories are used to interpreting the HCHO variability. More analysis of data and evidence is needed. There should be at least references to the rates of change of $CH_4$ and various NMVOCs reported in the literature over the same time periods.

**Response:** We have presented detailed statistics and analysis of the GEOS-Chem emission inventories to support the point of this paper in the revised version.

Specific comments for correction.

1. Abstract: The abstract should reflect that measurements from only one FTIR site (Hefei) were used for investigating the HCHO variability.

**Response:** Done!

2. Page 3: The first two paragraphs to be quite long while they bring very little useful

information to the manuscript, and hence they are not necessary. A concise description is suggested.

**Response:** Done!

3. Page6, line 24: YRD should be defined.

**Response:** Done!

4. Page 10, line 25: Where is the location of the CRDS analyzer? At the site where the $O_x$ in situ measurements is performed?

**Response:** In the revised version, we have stated that the CRDS analyser located side by side with the FTIR instrument. Please see section 2.1 and Fig. 1(b) for details.

5. Page 11, line 12: accounted

**Response:** Done!

6. Page11, line 33: Modelled HCHO was decreased by. . .

**Response:** Done!

7. Page11, line 37: As a short-lived species

**Response:** Done!

---

## Referee Report (RR1)

The manuscript by Sun et al. presents and analyses a 5-year time series of new HCHO measurements obtained from ground-based FTIR spectra recorded between 2015 and 2019 at Hefei, eastern China. Seasonal and interannual variabilities of HCHO were analysed and hydroxyl (OH) radical production rates from HCHO photolysis at the measurement site were also evaluated. A statistical model is adjusted in order to reproduce the HCHO abundance and variability based on ground level in situ measurements of CO and $O_x$ ($O_3$ + $NO_2$) taken in the vicinity of the FTIR site. CO and $O_x$ are used as tracers for emitted and photochemical HCHO, respectively, in order to estimate the contribution of direct emissions and oxidation of gas precursors to the HCHO abundance and variability. Finally, contributions of emission sources from various categories and geographical regions in China to the observed HCHO summertime enhancements were determined by using a series of GEOS-Chem sensitivity simulations.

I read the revised version, and went back and looked over the original version. The authors have put in a nice effort into revising their manuscript and mainly responding to the critiques of the original manuscript (especially by reviewer #1). The questionable investigation of diurnal modulation of HCHO at Hefei with GEOS-Chem simulations is removed. The discussion on the retrievals, kernels, uncertainties is well done. The revised version exploits the resulting new dataset extensively and presents sufficient perspectives in relation with the literature. The FTIR column-averaged dry air mole fractions of HCHO has been compared with the ground level measurements to ensure that the HCHO column measurements at Hefei can be used as representative of near-surface conditions. The interpretation of the GEOS-Chem results now has a much clearer message and storyline. The observed increasing change rate of HCHO from 2015 to 2019 over Hefei was attributed to the increase in photochemical HCHO resulting from increasing change rates of both $CH_4$ and NMVOCs oxidations, which overwhelmed the decrease in emitted HCHO. This study provides a valuable evaluation of recent VOCs emissions and regional photochemical capacity in China.

I think that this is now publishable in the ACP, after a few mostly minor revisions. Generally, the English should be improved throughout the manuscript, especially in the newly inserted sections or sentences, e.g., p17-L42, "drive" should be "driving", p18-L29, "Conclusions" is better than "Concluding remarks". I am not a native speaker, therefore I did not attempt to correct all these flaws throughout the whole paper. I assume that much of this will be done in the copy-editing phase. For the reference section, inconsistent order of first names and family names between different references are found, and some with DOI link, while some are not or with wrong DOI. The authors should refer to the guideline for reference in the ACP homepage. Nevertheless, this is interesting data from a highly polluted region that should be in the literature. Hence, I feel this should be published.

---

## Author Response (AR2)

**We thank the comments from the editor and two referees. Manuscript uploaded in this version has been subjected to technical correction. Please check the point-by-point response as follows.**

**(1) Detailed response to comments from referees:**

The manuscript by Sun et al. presents and analyses a 5 year time series of new HCHO measurements obtained from ground based FTIR spectra recorded between 2015 and 2019 at Hefei, eastern China. Seasonal and interannual variabilities of HCHO were analysed and hydroxyl (OH) radical production rates from HCHO photolysis at the measurement site were also evaluated. A statistical model is adjusted in order to reproduce the HCHO abundance and variability based on ground level in situ measurements of CO and $O_x$ ($O_3$+$NO_2$ ) taken in the vicinity of the FTIR site. CO and $O_x$ are used as tracers for emitted and photochemical HCHO, respectively, in order to estimate the contribution of direct emissions and oxidation of gas precursors to the HCHO abundance and variability. Finally, contributions of emission sources from various categories and geographical regions in China to the observed HCHO summertime enhancements were determined by using a series of GEOS-Chem sensitivity simulations.

I read the revised version, and went back and looked over the original version. The authors have put in a nice effort into revising their manuscript and mainly responding to the critiques of the original manuscript (especially by reviewer #1) The questionable investigation of diurnal modulation of HCHO at Hefei with GEOS-Chem simulations is removed . The discussion on the retrievals, kernels, uncertainties is well done. The revised version exploits the resulting new data set extensively and present s sufficient perspectives in relation with the literature. The FTIR column averaged dry air mole fractions of HCHO has been compared with the ground level measurements to ensure that the HCHO column measurements at Hefei can be used as representative of near surface conditions. The interpretation of the GEOS-Chem results now has a much clearer message and storyline. The observed increasing change rate of HCHO from 2015 to 2019 over Hefei was attributed to the increase in photochemical HCHO resulting from increasing change rates of both $CH_4$ and NMVOCs oxidations, which overwhelmed the decrease in emitted HCHO. This study provides a valuable evaluation of recent VOCs emissions and regional photochemical capacity in China.

I think that this is now publishable in the ACP, after a few mostly minor revisions. Generally, the English should be improved throughout the manuscript, especially in the newly inserted sections or sentences, e.g., p17 L42, "drive"

should be "driving", p18 L29, "Conclusions" is better than " Concluding remarks ". I am not a native speaker, therefore I did not attempt to correct all these flaws throughout the whole paper. I assume that much of this will be done in the copy editing phase. For the reference section, inconsistent order of first names and family names between different references are found, and some with DOI link, while some are not or with wrong DOI. The authors should refer to the guideline for reference in the ACP homepage. Nevertheless, this is interesting data from a highly polluted region that should be in the literature. Hence, I feel this should be published.

**Response:** This paper has been subjected to technical correction based on above comments. First, the language problem in p17 L42 and p18 L29 have been addressed. In addition, one of the coauthors with good command of English has copy-edited the rest parts. I assume that the copy editing phase will further improve the grammars. Second, the reference has been updated to follow the ACP guideline. Since all revisions are minor based technical correction. We did not marked up the revised paper.